# UniMapping: Unified SLAM Framework for Map-Centric Embodied Perception

**Xiaze Zhang**[1] **Ziheng Ding**[1] **Yuejie Zhang**[1] **Lifeng Chen**[1] **Rui Feng**[1 2]

## Abstract

Simultaneous Localization and Mapping (SLAM) is increasingly expected to provide reusable spatial representations for downstream perception. However, existing approaches often struggle with scale-consistency and producing maps that lack the geometric fidelity required for reliable perception. We propose *UniMapping*, a unified SLAM framework that constructs a persistent neural-descriptor map from multimodal observations. We introduce a **Spatial-Aware Deformable Transformer** that injects explicit geometric inductive bias to ensure scale-invariant feature extraction, alongside a **Spatial Fusion** strategy that decouples feature aggregation from temporal sequences. Extensive experiments on both indoor and outdoor benchmarks demonstrate competitive SLAM performance. Notably, our method significantly enhances downstream tasks (mAP $+3.1\%$ and mIoU $+6.4\%$) by leveraging accumulated multi-view context.

## 1. Introduction

Simultaneous Localization and Mapping (SLAM) aims to jointly estimate poses and reconstruct the environment. As embodied intelligence advances, the map is not merely a geometric abstraction but a perceptual foundation facilitating perception tasks. However, most existing approaches treat SLAM and perception as isolated problems, which leads to fragmented representations and redundant computation, limiting the efficiency across tasks and environments. Bridging localization and scene understanding through a unified map-centric representation that synergizes geometry and semantics within a shared spatial representation is a promising research direction.

[1]College of Computer Science and Artificial Intelligence, Shanghai Key Laboratory of Intelligent Information Processing, Fudan University, Shanghai, China [2]College of Intelligent Robotics and Advanced Manufacturing, Fudan University, Shanghai, China. Correspondence to: Rui Feng <fengrui@fudan.edu.cn>.

*Proceedings of the 43rd International Conference on Machine Learning*, Seoul, South Korea. PMLR 306, 2026. Copyright 2026 by the author(s).

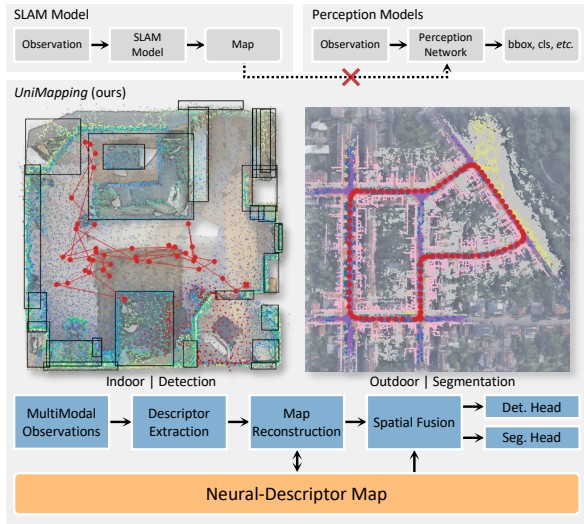

*Figure 1. UniMapping* architecture reconstructs a consistent global neural-descriptor map that supports diverse environmental scales, and downstream perception tasks with a unified framework.

*One fundamental challenge is how to extract features with both geometric structure and semantic cues consistently across diverse scenes.* Geometry-based SLAMs (Zhang & Singh, 2014; Chen et al., 2019; Dellenbach et al., 2022; Vizzo et al., 2021; 2023) represent scenes using fundamental geometric elements. They prioritize efficiency and geometric consistency but offer limited support for semantic understanding. Neural implicit representations (Sucar et al., 2021; Zhu et al., 2022; Johari et al., 2023; Liso et al., 2024) provide expressive continuous scene models, yet their optimization-centric parameterization makes them difficult to reuse or adapt to large-scale scenarios. Descriptor-based representations (Dube et al., 2020; Zhang et al., 2024b;a) represent the scene with learned spatial descriptors. However, existing systems mainly focus on geometric matching, limiting their ability to capture spatial context across observations and support downstream perception.

*Another challenge is how to aggregate multi-frame features to provide a complete spatial representation for perception tasks.* Most perception approaches (Shen et al., 2024; Feng et al., 2024) operate in a single-frame manner, extracting features independently from each observation, thereby limiting cross-view consistency. Temporal aggregation strategies (Hu et al., 2023b; Wang et al., 2023b; He et al., 2023;

Li et al., 2024; Antunes et al., 2024) extend this paradigm by introducing temporal memory bank, but their computational and memory costs typically grow with the sequence length. In contrast, while temporal observations are inherently unbounded, the spatial extent of a scene is usually finite and structured. Ideally, SLAM should bridge these domains by converting unbounded temporal observations into a structured, finite spatial representation. This suggests that organizing and aggregating features directly in the spatial domain can provide a theoretically more efficient and reusable representation for perception.

To address these challenges, we propose *UniMapping*, a unified SLAM framework that integrates multimodal observations into a consistent spatial neural-descriptor map, enabling both accurate mapping and direct support for downstream perception tasks as illustrated in Figure 1. We introduced the **Spatial-Aware Deformable Transformer** to extract neural-descriptors. Unlike standard attention mechanisms that struggle with scale ambiguity in projective geometry, we inject an explicit geometric inductive bias into the feature extraction process to ensure scale-invariance by design. Furthermore, neural-descriptors are organized via a spatial index, addressing issues of memorization efficiency and long-term consistency of temporal-based framework. Our proposed **Spatial Fusion** mechanism aggregates multi-view information online, allowing perception tasks to reuse spatially accumulated features without redundant feature extraction from raw observations.

In contrast to decoupled SLAM and perception pipelines, *UniMapping* constructs a shared neural-descriptor map that is simultaneously reusable for localization and perception, which avoids redundant feature extraction, preserves long-term spatial context, and establishes geometry-semantic consistency within a single framework. *UniMapping* serves as a persistent spatial foundation for embodied perception.

**Our contributions can be summarized as:**

- We present a unified SLAM framework, *UniMapping* that bridges the gap between geometric SLAM and semantic perception. It generalizes across environments solely through learned neural-descriptors.

- We introduce the **Spatial-Aware Deformable Transformer**, which leverages geometric priors to achieve scale-invariant feature extraction, and **Spatial Fusion** to enhance multi-frame feature aggregation, improving downstream perception performance.

- We demonstrate that our map-centric perception paradigm not only achieves competitive SLAM accuracy but also significantly boosts downstream tasks by leveraging spatially fused long-term context.

## 2. Related Work

### 2.1. Feature Extraction and Scene Representation

A challenge in SLAM is to extract representative features while remaining adaptable to diverse sensors and scenes.

**Photometric-** or **Geometry-Based** approaches construct maps using low-level elements. Visual SLAM (Mur-Artal & Tardós, 2017) rely on reprojection error and LiDAR-based methods (Zhang & Singh, 2014; Dellenbach et al., 2022; Vizzo et al., 2021; 2023) focus on geometric consistency. They achieve high efficiency but lack geometric understanding. Hybrid visual–LiDAR systems (Shan et al., 2020; Huang et al., 2021; Kang et al., 2026) improve robustness through multi-sensor integration, yet their reconstructed geometric maps still lack perceptual features. **NeRF-Based** or **Neural Implicit** models (Sucar et al., 2021; Zhu et al., 2022; Johari et al., 2023; Rosinol et al., 2023; Hu et al., 2023a) represent scenes as continuous functions learned via neural networks, which are difficult to scale to large, unbounded environments. Meanwhile, the learned features are not readily accessible for downstream perception tasks. **Descriptor-Based** methods aim to encode features as local descriptors. *DeepPointMap*s (Zhang et al., 2024b;a) pioneered neural descriptors for SLAM, demonstrating their effectiveness for localization. However, these work primarily focused on outdoor driving scenarios, and the descriptors were mainly optimized for localization rather than serving as reusable representations for perception tasks. Recently, **3DGS-Based** methods (Keetha et al., 2024; Yugay et al., 2023; Hu et al., 2024) leverage explicit Gaussian fields for high-fidelity mapping, yet the millions of kernels often incur prohibitive memory overhead and structural gaps in unobserved regions.

In summary, existing representations tend to emphasize either photometric fidelity or local geometric matching, but struggle to simultaneously achieve multimodal integration, scalability, and perception-oriented feature expressiveness.

### 2.2. SLAM for Perception Tasks

Downstream perception tasks in embodied intelligence, *e.g.*, 3D object detection and semantic segmentation, benefit from aggregating multi-frame observations to address occlusions and viewpoint changes.

**Single-Frame** perception methods (Shen et al., 2024; Feng et al., 2024; Wu et al., 2024; Liang et al., 2024) operate independently on each observation, providing low-latency and computationally efficient predictions. However, they lack cross-view consistency and cannot accumulate information from past observations, leading to unstable predictions under occlusion or abrupt viewpoint changes. **Temporal-Based** methods (Hu et al., 2023b; Wang et al., 2023b; He et al., 2023; Li et al., 2024; Antunes et al., 2024) exploit

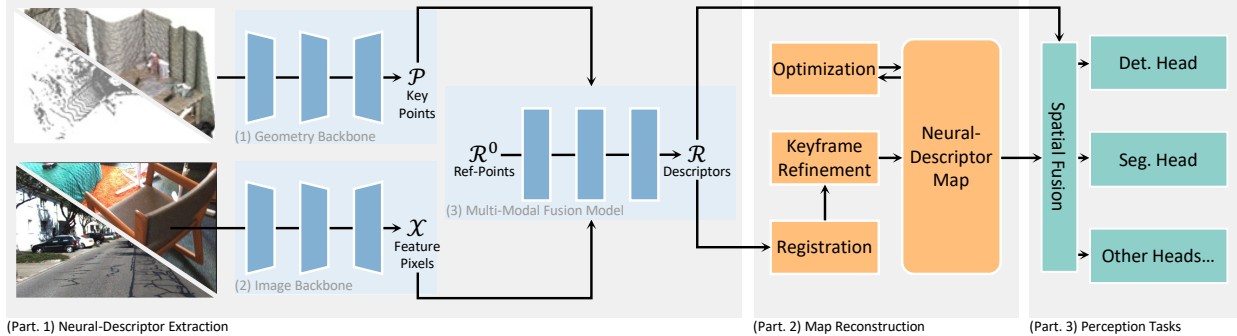

*Figure 2.* **The Overall Architecture of *UniMapping*.** Our approach takes observation sequence as input for pose estimation, map reconstruction and downstream perception tasks.

sequential observations to improve stability and capture environmental dynamics. They usually use memory banks or RNNs to aggregate features across time. While these approaches partially leverage multi-frame information, they suffer from weak spatial alignment, as features are often aggregated through weakly aligned cross-attention mechanisms without explicit spatial anchoring. Consequently, temporal fusion introduces spatial inconsistency, memory growth, and difficulties in querying long-term observations, particularly for large-scale or looped trajectories. **Mapping-Based** approaches (Dube et al., 2020; Matsuki et al., 2024) leverage pose alignment or map-level representations to anchor observations in space. Despite improving spatial consistency, perception features are typically task-specific and not maintained as a unified spatial representation.

In summary, effectively integrating perception with spatial reasoning remains challenging, as existing methods lack spatially aligned feature representations for aggregating long-term observations into reusable maps.

## 3. Method

As illustrated in Figure 2, *UniMapping* is a multimodal SLAM architecture containing three parts: (1) **Neural-Descriptor Extraction** model extracts features from multimodal sensors and encodes neural-descriptors, (2) **Map Reconstruction** module constructs a consistent neural-descriptor map, and (3) **Perception Tasks** part aggregates the neural-descriptors with spatial information and feeds them to subsequent heads to accomplish perception tasks.

### 3.1. Neural-Descriptor Extraction

*UniMapping* utilizes *neural-descriptors* $\mathcal{R}$ to represent 3D scenes. A neural-descriptor $r \in \mathcal{R}$ can be represented as $r = \{r^{\mathrm{xyz}}, r^{\mathrm{feat}}\}$, where $r^{\mathrm{xyz}}$ denotes its 3D coordinate and $r^{\mathrm{feat}}$ is its feature. The extraction model aims to extract neural-descriptors $\mathcal{R}$ from multimodal sensor inputs (*i.e.*, image $\mathcal{I}$ and point cloud $\mathcal{P}$). As illustrated in Figure 3, the

extraction module consists of three parts.

**Image Encoder** and **Geometry Encoder** extracts multiscale feature-pixels $\mathcal{X}_i$ (voxel-based features $\mathcal{V}_i$) from given image $\mathcal{I}$ (point cloud $\mathcal{P}$). We utilize ConvNeXt (Liu et al., 2022) with FPN (Lin et al., 2017) as our image backbone and 3D MinkowskiNet (Choy et al., 2019) as geometric backbone, respectively. For each layer $l$, We denote the feature-pixel located at pixel coordinate $(u, v)$ as $x^{uv} \in \mathcal{X}_l$ and feature-voxels at coordinate $(x, y, z)$ as $v^{xyz} \in \mathcal{V}_l$.

**Multimodal Fusion Model** integrates the aforementioned multimodal cues and generates neural-descriptors $\mathcal{R}$, which consists of a Reference Point Generator followed by multiple Spatial-Aware Deformable Transformer (SADT) layers.

The Reference Point Generator samples a set of spatial coordinates (*i.e.*, *reference-points* $r_0 \in \mathcal{R}_0$) with farthest-point-sample on the input point cloud to obtain reference coordinates $r_0^{\mathrm{xyz}}$, and initialize its feature with learnable positional embedding $r_0^{\mathrm{feat}} = \mathrm{MLP}\left(r_0^{\mathrm{xyz}}\right)$. $\mathcal{R}_0$ are further progressively refined across multiple SADT layers without changing their spatial coordinates. As shown in Figure 3 (3), each SADT layer aggregates information from multimodal representations using Image branch (3.1) and Geometric branch (3.2). After gathering multimodal cues, the fusion network (3.3) fuses those multimodal features.

In image branch, given a reference descriptor $r_i = \{r_i^{\mathrm{xyz}}, r_i^{\mathrm{feat}}\}$ in the local coordinate system, we project its 3D location onto the image plane using camera extrinsics $[R_{\mathrm{cam}}|T_{\mathrm{cam}}]$ and intrinsics $K_{\mathrm{cam}}$

$$r_i^{\mathrm{c}} = R_{\mathrm{cam}} r_i^{\mathrm{xyz}} + T_{\mathrm{cam}}, \; r_i^{\mathrm{d}} = (r_i^{\mathrm{c}})_z, \; r_i^{\mathrm{uv}} = \pi(K_{\mathrm{cam}} r_i^{\mathrm{c}}) \quad (1)$$

where $\pi(\cdot)$ denotes perspective division. Directly sampling features at $r_i^{\mathrm{uv}}$ often fails to capture accurate contextual appearance due to scale uncertainty introduced by perspective projection. Inspired by deformable attention (Zhu et al., 2020), we adaptively aggregate contextual visual information. For each reference point $r_i$, the network predicts

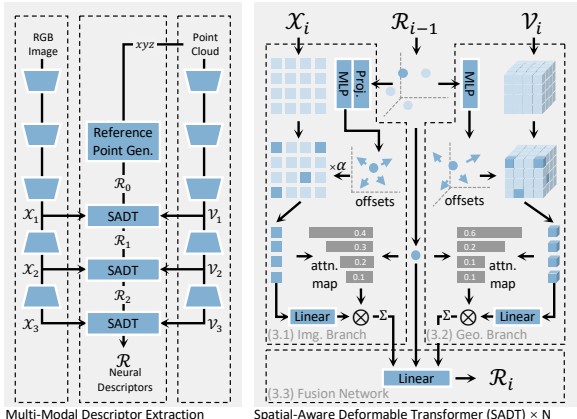

*Figure 3.* **Overview of Neural-Descriptor Extraction Model**. The model takes image and point cloud as inputs, and fuses the multi-scale feature tokens into neural-descriptors using Spatial-Aware Deformable Transformer (SADT).

$N_{\text{offset}}^{\text{img}}$ *spatial-aware learnable offsets* as

$$\Delta r_{i,k}^{\text{uv}} = \alpha \cdot s_{\text{img}} \cdot \tanh\left(\text{MLP}\left(r_i^{\text{feat}}\right)\right) \quad (2)$$

where $k = 1, \ldots, N_{\text{offset}}^{\text{img}}$ indexes each sampling location and $s_{\text{img}}$ is a predefined scale factor.

To bridge the 2D-3D geometric gap, SADT introduces a *perspective-aware scaling factor* $\alpha = 1/r_i^{\text{d}}$. Based on pin-hole geometry, this factor normalizes the learnable offsets $\Delta r_{i,k}^{\text{uv}}$ to ensure a constant physical receptive field across varying depths. By modulating sampling offsets with inverse depth, SADT achieves scale-invariant feature extraction, effectively neutralizing perspective distortions.

We then compute the sampled 2D features with bilinear sampling around these selected locations and utilize weight multi-head fusion to obtain the aggregated feature $r_i^{\text{img}}$ as

$$r_i^{\text{img}} = \sum_{h=1}^{H} \sum_{k=1}^{N_{\text{offset}}^{\text{img}}} A_{i,h,k}^{\text{img}} \, \phi\left(r_i^{\text{uv}} + \Delta r_{i,k}^{\text{uv}}, \mathcal{X}_i\right) \quad (3)$$

where $\phi_2(\cdot)$ is bilinear sampling applied on feature-pixel $\mathcal{X}$ and $A_{i,h,k}^{\text{img}}$ are attention weights learned jointly with the refinement network.

For Geometry Branch, given a reference descriptor $r_i = \{r_i^{\text{xyz}}, r_i^{\text{feat}}\}$, our goal is to gather informative 3D features from its local geometric neighborhood. Unlike neighborhood-based operators (*e.g.*, sparse conv. (Thomas et al., 2019) and set-abstract (Qi et al., 2017)) that rely on predefined geometric extents, we learn a set of query-dependent 3D offsets to explore geometric regions dynamically. Specifically, the module predicts $N_{\text{offset}}^{\text{geo}}$ offsets in the 3D space similar to Equation (3)

$$\Delta r_{i,k}^{\text{xyz}} = s_{\text{geo}} \cdot \tanh\left(\text{MLP}\left(r_i^{\text{feat}}\right)\right) \quad (4)$$

where $k = 1, \ldots, N_{\text{offset}}^{\text{geo}}$ and static parameter $s_{\text{geo}}$ controls the geometric search range. To efficiently locate relevant voxels in large-scale maps, we use a hash-based neighbor query over $\mathcal{V}$ to retrieve geometric features surrounding. We then apply trilinear interpolation to obtain smooth geometric responses, suppressing discretization artifacts from voxelization. The aggregated geometric feature is obtained by multi-head deformable attention

$$r_i^{\text{geo}} = \sum_{h=1}^{H} \sum_{k=1}^{N_{\text{offset}}^{\text{geo}}} A_{i,h,k}^{\text{geo}} \, \phi\left(r_i^{\text{xyz}} + \Delta r_{i,k}^{\text{xyz}}, \mathcal{V}_i\right) \quad (5)$$

where $\phi_3(\cdot)$ extracts voxel-aligned features and $A_{i,h,k}^{\text{geo}}$ are attention weights. This design enables reference descriptors to attend to geometrically informative structures adaptively. Moreover, since offsets are learned relative to descriptors rather than an absolute scale, the method preserves strong robustness and generalization across diverse indoor and outdoor environments.

The Fusion Network finally generates the final feature of the descriptor by fusing appearance and geometry cues with a learnable approach

$$r_{i+1}^{\text{feat}} = \text{FFN}\left(r_i^{\text{feat}} + r_i^{\text{img}} + r_i^{\text{geo}}\right) \quad (6)$$

where FFN is an MLP incorporating modality-specific cues, ensuring both semantic richness and geometric consistency.

### 3.2. Map Reconstruction

To incrementally maintain the neural-descriptor map, we adapt the mature registration and mapping framework following Zhang et al. (2024a) and further optimize its robustness and memory consumption.

**Registration**. The Registration Network is used to solve the rigid transformation between two sets of descriptors $\mathcal{R}_1$ and $\mathcal{R}_2$. The registration network first exchanges the information between them using a cross-attention block, and the *pairing network* $\text{H}_{\text{pair}}$ estimates the correspondence $\sigma_{ij}$ between descriptors ($r_i \in \mathcal{R}_1$ and $r_j \in \mathcal{R}_2$) based on pairwise feature similarity $\odot$ as $\sigma_{ij} = \text{H}_{\text{pair}}\left(r_i^{\text{feat}}\right) \odot \text{H}_{\text{pair}}\left(r_j^{\text{feat}}\right)$. The confidence $\varepsilon$ of this registration is calculated as the average similarity over all pairs. To overcome the sparseness of descriptors, an additional *offset network* $\text{H}_{\text{offset}}$ predicts the relative offsets $\delta_{ij}$ between descriptor pairs $r_i$ and $r_j$ as $\delta_{ij} = \text{H}_{\text{offset}}\left(r_i^{\text{feat}} \oplus r_j^{\text{feat}}\right)$, where $\oplus$ denotes concatenate.

Based on the weighted correspondences and refined point coordinates, we estimate the relative transformation $R, T$ by solving a weighted orthogonal procrustes problem Schönemann (1966) as

$$\arg\min_{R,T} \sum_{ij} \omega_{ij} \|(R \times (r_i^{\text{xyz}} + \delta_{ij}) + T - r_j^{\text{xyz}})\|_2^2 \quad (7)$$

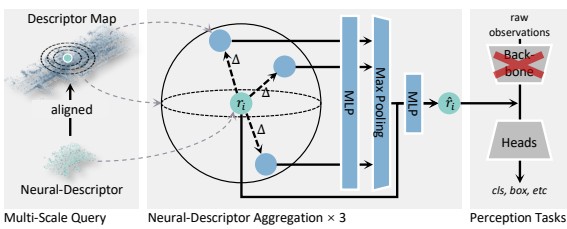

*Figure 4.* **Overview of the Spatial Fusion Module**, which aggregates multi-view neural descriptors from the persistent map into current observations through spatially-guided local fusion.

To improve robustness against outliers and noisy correspondences, we employ the Geman-McClure robust kernel for adaptive weighting. Specifically, the correspondence weight is defined as

$$\omega_{ij} = \frac{1}{1 + e_{ij}^2/\sigma^2} \qquad (8)$$

where $e_{ij}$ denotes the residual error and $\sigma$ is a robust scale parameter estimated from the median residual.

**Mapping**. We utilize pose-graph to store the reconstructed neural-descriptor map, where frames are *nodes* and estimated transformations are *edges*. When the registration confidence $\varepsilon$ exceeds $\varepsilon_{\text{odom}}$, an *odom-edge* is added into pose-graph. A key-frame selection process is applied to determine if the frame should be assigned as a keyframe and refined with frame-to-map refinement procedure to improve the pose estimation accuracy. To maintain global consistency, loop-closure detection and pose-graph optimization (PGO) are performed. For each keyframe, previous scans are treated as loop candidates and filtered by field-of-view consistency. A *loop-edge* is added if the registration confidence exceeds $\varepsilon_{\text{loop}}$. Once a loop is confirmed, a global optimization will be triggered. *Detailed mapping procedures are provided in appendix B.*

### 3.3. Perception Tasks

*UniMapping* transforms the time-series observation into a spatial-indexed neural-descriptor map. Downstream perception tasks can directly leverage the map to achieve more accurate predictions.

**Spatial Fusion.** Given the neural-descriptor map and the estimated pose, we introduce *Spatial Fusion* module that bridges the neural-descriptor map and task-specific prediction heads. For each query descriptor $r_i \in \mathcal{R}_i$ from the current observation, we apply a multi-scale neural-descriptor aggregation to produce the Spatial Aggregated Descriptors $\hat{r}_i \in \tilde{\mathcal{R}}$, as illustrated in Figure 4. To capture local geometric structure while maintaining permutation invariance, a ball search retrieves a fixed number of neighbors $r_j \in \mathcal{R}_j$ within multiple Euclidean radius for each query. Each neighbor is encoded by concatenating its feature $r_j^{\text{feat}}$ with the relative offset $\Delta = r_j^{\text{xyz}} - r_i^{\text{xyz}}$, followed by a shared MLP and max-pooling. This query operation exploits the estimated pose to

establish spatial correspondence, avoiding any global soft attention and reducing computational overhead. Additionally, as Spatial Fusion employs the asymmetric fusion strategy, it can explicitly distinguish descriptors from the current observation from those stored in the map, thereby mitigating ghosting artifacts caused by moving objects.

**Perception Heads.** 3D perception models can be formulated as a mapping from observations to task-specific predictions, which can be formulated as

$$\mathcal{Y} = \text{H}_{\text{task}}(\text{N}(\mathcal{P}, \mathcal{X})) \qquad (9)$$

where N denotes the backbone producing latent context and $\text{H}_{\text{task}}$ is the task-specific prediction head predicting the final output. As shown in Figure 4, since *UniMapping* provides Spatial Aggregated Descriptors $\hat{\mathcal{R}}$ that encodes multi-view observations from map, the backbone can be replaced and perception tasks can be directly formulated as

$$\mathcal{Y} = \text{H}_{\text{task}}(\hat{\mathcal{R}}) \qquad (10)$$

For example, $\hat{\mathcal{R}}$ can serve as spatial queries in object detection tasks, or be injected at the bottleneck of semantic segmentation models as auxiliary representations. This design enables efficient reuse of the neural-descriptor map across multiple perception tasks without re-extracting features from raw sensor inputs while providing efficient features, which is critical for embodied perception. *Detailed task implementations are provided in Section 4.2.*

### 3.4. Training

**Descriptor Extraction**. We train the extraction network using InfoNCE loss (He et al., 2020) over $\mathcal{R}$ within and across frames. Given two descriptors $r_i$ and $r_j$, the pair is marked as a *positive pair* if $r_j$ is the nearest neighbor of $r_i$ and their distance $d_{ij}$ is below a threshold $d_0$. If $d_{ij} > d_0$, the pair is marked as *negative*. All remaining pairs are treated as *neutral* samples and excluded from the loss calculation. For each positive descriptor $r_i$, the contrastive loss is applied as

$$\mathcal{L}_{\text{c}} = -\log \frac{\exp\left((r_i \odot r_j)/\tau\right)}{\sum\limits_{k \in \text{neg}} \exp\left((r_i \odot r_k)/\tau\right)} \qquad (11)$$

where neg are the negative pairs of $r_i$, $\odot$ is cosine similarity, and $\tau$ is a temperature factor.

**Map Reconstruction**. We train the *Registration Network* with offset loss $\mathcal{L}_{\text{o}}$, which supervise the predicted $\delta_{ij}$ as

$$\mathcal{L}_{\text{o}} = \mathbb{E}_i \mathbb{E}_{j \in \text{pos}} \| R_{ij} \left( r_i^{\text{xyz}} + \delta_{ij} \right) + T_{ij} - r_i^{\text{xyz}} \|_{\Sigma} \qquad (12)$$

where pos is the positive pairs of $r_i$, the relative pose between frame $i$ to $j$ are denoted as $R, T$, and $\| \cdot \|_{\Sigma}$ represents the Mahalanobis distance.

*Table 1.* **Localization Accuracy on KITTI Odometry Benchmark (Trans↓ and Rot↓).**

| Type | Method | 06 | | 07 | | 08 | | 09 | | 10 | |
|---|---|---|---|---|---|---|---|---|---|---|---|
| | | Trans | Rot | Trans | Rot | Trans | Rot | Trans | Rot | Trans | Rot |
| **Geometric** | LOAM[L] | 0.65 | - | 0.63 | - | 1.12 | - | 0.77 | - | 0.79 | - |
| | LiLO[L] | 0.54 | 0.32 | 0.60 | 0.61 | 1.07 | 0.41 | 0.63 | 0.32 | 0.99 | 0.33 |
| | ORB-SLAM2[C] | 0.89 | 0.27 | 0.89 | 0.50 | 1.03 | 0.31 | 0.86 | 0.25 | 0.62 | 0.29 |
| | SOFT2[C] | 0.60 | 0.23 | 0.45 | 0.29 | 0.91 | 0.26 | 0.75 | **0.22** | 0.74 | **0.24** |
| | DEMO[LC] | 0.96 | - | 1.16 | - | 1.24 | - | 1.17 | - | 1.14 | - |
| | LAMV-SLAM[LC] | 0.49 | - | 0.84 | - | 1.19 | - | 0.80 | - | **0.55** | - |
| | Fang et al. (2023)[LC] | 0.55 | - | 0.66 | - | 0.96 | - | **0.60** | - | 0.51 | - |
| | NALO-VOM[LC] | 1.33 | - | 1.59 | - | 0.90 | - | 1.02 | - | 0.85 | - |
| | DVLO[LC] | 0.33 | **0.17** | 0.46 | 0.33 | 1.09 | 0.44 | 0.85 | 0.36 | 0.88 | 0.46 |
| | DVLO4D[LC] | **0.32** | 0.21 | 0.43 | 0.32 | 0.95 | 0.36 | 0.77 | 0.33 | 0.76 | 0.46 |
| **Descriptor** | LO-Net[L] | - | - | 0.56 | 0.45 | 1.08 | 0.43 | 0.77 | 0.38 | 0.92 | 0.41 |
| | DELO[L] | 0.83 | 0.35 | 0.58 | 0.41 | 1.36 | 0.64 | 1.23 | 0.57 | 1.53 | 0.90 |
| | TransLO[L] | - | - | 0.55 | 0.43 | 1.29 | 0.50 | 0.95 | 0.46 | 1.18 | 0.61 |
| | DeepPointMap[L] | 0.77 | - | 0.44 | - | 1.09 | - | 0.95 | - | 0.69 | - |
| | DeepPointMap2[LC] | 0.47 | 0.20 | 0.39 | **0.25** | 0.77 | **0.22** | 0.62 | 0.23 | 0.75 | 0.40 |
| | DiffLO[L] | - | - | 0.37 | 0.27 | 1.12 | 0.44 | 0.68 | 0.28 | 0.66 | 0.32 |
| | *UniMapping* [LC] | 0.44 | 0.20 | **0.38** | 0.28 | **0.73** | 0.42 | 0.66 | **0.22** | 0.91 | 0.40 |

Superscript represents the input modal for each approach: LiDAR modal[L], Camera modal[C], and LiDAR+Camera modal[LC].

**Perception Tasks**. We jointly train the perception heads with the networks mentioned above. Since *UniMapping* is not tailored to any specific perception head, we utilize their original loss for each task without modification. We adopt a sequential training strategy that organizes data into spatially coherent *clips*, enabling the model to learn spatially consistent representations from overlapping observations while maintaining training diversity (see *appendix* D).

## 4. Experiments

**Datasets.** We evaluate our method on two multimodal benchmarks covering both outdoor and indoor scenes. SemanticKITTI (Behley et al., 2019) contains 11 outdoor sequences ranging from urban to highway environments, which are split into 6 training sequences and 5 evaluation sequences. ScanNet (Dai et al., 2017) is an indoor dataset comprising 1513 RGB-D sequences captured with handheld devices, which contains 5 sequences for SLAM validation following (Sucar et al., 2021), and 300 sequences for detection evaluation, with all remaining data used for training.
**Metrics.** For SLAM evaluation, we follow the official metrics: Relative Translation Error (Trans↓) (m/100m) and Average Rotation Error (Rot↓) (°/100m) for KITTI (Geiger et al., 2012), and RMSE Absolute Translation Error (ATE↓) (cm) for ScanNet (Dai et al., 2017). For perception tasks, we report mAP/mAR↑ for object detection (Shen et al., 2024), and mIoU↑ for semantic segmentation (Tang et al., 2020).
**Training.** *UniMapping* is trained for 20 epochs on both datasets, using 4×RTX 3090 GPUs with AdamW (Reddi et al., 2019) and Muon (Jordan et al., 2024) optimizers, with

a cosine learning rate schedule ($lr = 1e - 3$). *Detailed training approach is provided in appendix D.*

### 4.1. Localization Accuracy

We evaluate the localization accuracy of *UniMapping* across large-scale outdoor scenarios and complex indoor scenes. For the outdoor scenario, we conduct the experiment on SemanticKITTI benchmark with LiDAR-Camera input. *UniMapping* is compared with state-of-the-art SLAM approaches including geometric-based LOAM (Zhang & Singh, 2017), LiLO (Velasco-Sánchez et al., 2023), ORB-SLAM2 (Mur-Artal & Tardós, 2017), SOFT2 (Cvišić et al., 2022), DEMO (Zhang et al., 2017), LAMV-SLAM (Yin et al., 2022), Fang et al. (2023), NALO-VOM (Hu et al., 2023c), DVLO (Liu et al., 2024), and DVLO4D (Liu et al., 2025), as well as the descriptor-based approaches LO-Net (Li et al., 2019), DELO (Ali et al., 2023), TransLO (Liu et al., 2023), DeepPointMap (Zhang et al., 2024b), Deep-PointMap2 (Zhang et al., 2024a) and DiffLO (Huang et al., 2025). Table 1 report the outdoor localization accuracy. Our method demonstrates competitive performance.

We also conduct indoor experiments on 5 scenes from the ScanNet benchmark using RGB-D input. *UniMapping* is compared with advanced indoor SLAM approaches including (a) implicit methods GO-SLAM (Yang et al., 2015), iMAP (Sucar et al., 2021), NICE-SLAM (Zhu et al., 2022), Vox-Fusion (Yang et al., 2022), ESLAM (Johari et al., 2023), MIPS-Fusion (Tang et al., 2023), and Loopy-SLAM (Liso et al., 2024), and (b) explicit descriptor- and gs-based Co-SLAM (Wang et al., 2023a), SplaTAM (Keetha et al., 2024),

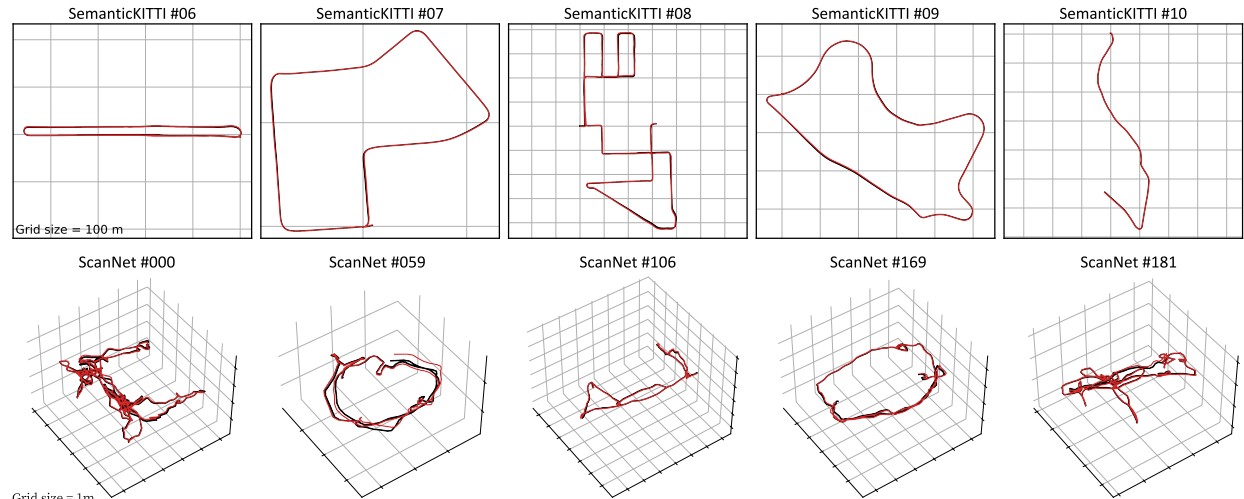

*Figure 5.* **Estimated Trajectories in Both Indoor and Outdoor Scenarios**. The black and red lines represent ground-truth and estimated trajectories. Our proposed *UniMapping* successfully achieves accurate localization in both large-scale and small-scale scenarios.

*Table 2.* **Localization Accuracy on ScanNet (ATE↓).**

| Type | Method | 000 | 059 | 106 | 169 | 181 |
|---|---|---|---|---|---|---|
| **Implicit** | GO-SLAM | 5.4 | 7.5 | 7.0 | 7.7 | 6.8 |
| | iMAP | 42.7 | 17.8 | 15.0 | 39.1 | 24.7 |
| | NICE-SLAM | 12.0 | 14.0 | 7.9 | 10.9 | 13.4 |
| | Vox-Fusion | 16.6 | 24.2 | 8.4 | 27.3 | 23.3 |
| | ESLAM | 7.3 | 8.5 | 7.5 | **6.5** | 9.0 |
| | MIPS-Fusion | 7.9 | 10.7 | 9.7 | 9.7 | 14.2 |
| | Loopy-SLAM | **4.2** | 7.5 | 8.3 | 7.5 | 10.6 |
| **Explicit** | Co-SLAM | 7.2 | 12.3 | 9.6 | 6.6 | 13.4 |
| | SplaTAM | 12.8 | 10.1 | 17.7 | 12.1 | 11.1 |
| | Gaussian-SLAM | 21.2 | 12.8 | 13.5 | 16.3 | 21.0 |
| | CG-SLAM | 7.1 | 7.5 | 8.9 | 8.2 | 11.6 |
| | MonoGS | 9.8 | 32.1 | 8.9 | 10.7 | 21.8 |
| | LoopSplat | 6.2 | **7.1** | 7.4 | 10.6 | 8.5 |
| | *UniMapping* | 13.7 | 7.4 | **5.7** | 7.7 | **3.7** |

*Table 3.* **Detection Performance on ScanNet Benchmark (AP↑ and AR↑)**. Details reported in *appendix* E.2.

| Method | $mAP_{25}$ | $mAP_{50}$ | $mAR_{25}$ | $mAR_{50}$ |
|---|---|---|---|---|
| V-DETR | 57.4 | 39.1 | 76.0 | 56.2 |
| ours (*w/o* $S_{patial}$ $F_{usion}$) | 57.7 | 36.6 | 75.9 | 52.7 |
| ours (*w/* $T_{emporal}$ $F_{usion}$) | 59.2 | 39.0 | 77.0 | 55.1 |
| ours (*w/* $S_{patial}$ $F_{usion}$) | **60.8** | **40.4** | **78.2** | **56.6** |

*Table 4.* **Segmentation Performance on Semantic-KITTI Benchmark (IoU↑)**. Details reported in *appendix* E.3.

| Method | mIoU | car | bicyclist | person |
|---|---|---|---|---|
| SPVCNN | 50.7 | **91.0** | 41.3 | 40.7 |
| ours (*w/o* $S_{patial}$ $F_{usion}$) | 53.7 | 89.5 | 65.2 | 44.6 |
| ours (*w/* $T_{emporal}$ $F_{usion}$) | 53.8 | 89.6 | 65.9 | 44.5 |
| ours (*w/* $S_{patial}$ $F_{usion}$) | **60.3** | 90.2 | **76.1** | **64.2** |

Gaussian-SLAM (Yugay et al., 2023), CG-SLAM (Hu et al., 2024), MonoGS (Matsuki et al., 2024), and LoopSplat (Zhu et al., 2025). As shown in Table 2, our method outperforms the comparisons on 2 sequences, without introducing any indoor-specific architectural components. We also evaluate our approach on an additional 300 sequences from the *val* set. Our approach achieves an ATE of $3.1^{+5.2}_{-1.1}$ cm, suggesting that positioning errors remain below $8.3$ cm in over 84% of indoor scenarios. As shown in Figure 5, our proposed *UniMapping* achieves accurate localization in both environments.

Due to the strong specialization of existing methods to specific environments, only a limited number can be consistently evaluated across both scenarios. Despite this, *UniMapping* achieves competitive performance in both environments by leveraging a unified neural-descriptor extraction framework that produces scale-invariant features. By relying on a consistent architecture, our method not only provides a reliable SLAM backbone but also establishes a

generalizable foundation for downstream perception tasks without requiring task- or environment-specific redesign.

### 4.2. Perception Performance

The core of *UniMapping* lies in bridging the SLAM task with downstream perception. We integrate two perception tasks into *UniMapping* with minimal task-specific adaptation to demonstrate the generality and reusability of the neural-descriptor map.

**Object Detection**. We first investigate object detection on ScanNet dataset, which requires high-level reasoning under complex environments. We adapt V-DETR (Shen et al., 2024) as baseline and train it with frame-wise inputs. We also attach the V-DETR detection head to our *UniMapping*, where Spatial Aggregated Descriptors $\hat{\mathcal{R}}$ is provided as input. Results are shown in Table 3. Based on the estimated poses, our model without Spatial Fusion achieves comparable performance to the V-DETR baseline, validating

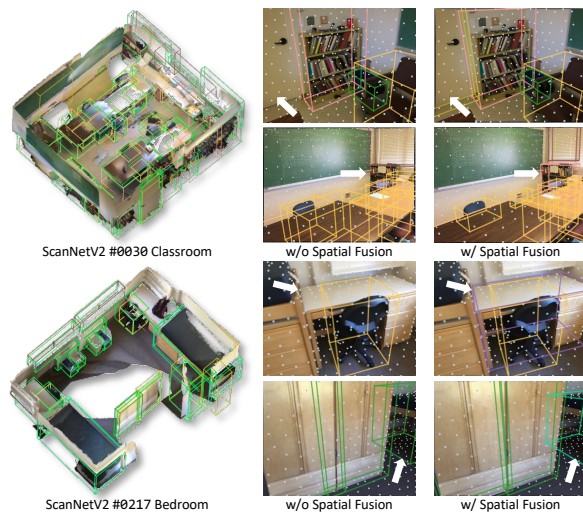

*Figure 6.* **Visualization of Detection**. The performance of occluded objects is improved. Arrows indicate instances of false negatives and mis-classification, and dots indicate neural-descriptors.

*Table 5.* **Ablation on Multimodal Fusion Model (ATE↓).**

| Method | 000 | 059 | 106 | 169 | 181 |
|---|---|---|---|---|---|
| Cross Attention | 21.6 | 27.4 | 11.4 | 22.9 | 18.0 |
| Interpolation | 14.0 | 15.8 | 7.6 | 21.1 | 12.9 |
| Deformable Attn. | 18.9 | 27.1 | 9.6 | 20.9 | 16.6 |
| SADT (ours) | **13.4** | **14.9** | **7.3** | **19.1** | **12.4** |

*Table 6.* **Ablation on Spatial Fusion Strategy (AP↑ and AR↑).**

| Method | mAP$_{25}$ | mAP$_{50}$ | mAR$_{25}$ | mAR$_{50}$ |
|---|---|---|---|---|
| w/o Spatial Fusion | 44.09 | 27.25 | 67.27 | 41.99 |
| Cross Attention | 41.29 | 25.34 | 63.00 | 39.38 |
| Deformable Attn. | 46.27 | 28.05 | 66.68 | 42.53 |
| Spatial Fusion (ours) | **48.84** | **30.62** | **69.12** | **45.86** |

that the learned descriptors inherently encode rich object-level semantics. Enabling Spatial Fusion yields a significant +3.4% mAP$_{25}$ improvement over V-DETR. Spatial Fusion outperforms the Temporal Fusion variant (*i.e.*, limited to a 10-frame window), demonstrating that a persistent, globally-consistent map provides superior context compared to transient temporal buffers. Illustrated in Figure 6, by retrieving global context from the neural-descriptor map, *UniMapping* effectively suppresses false negatives and mis-classifications arising from incomplete viewpoints, establishing the map as a reusable perceptual foundation.

**Semantic Segmentation**. We evaluate dense prediction on SemanticKITTI by integrating the SPVCNN (Tang et al., 2020) decoder into *UniMapping*. As shown in Table 4, *w/o* Spatial Fusion outperforms the SPVCNN baseline, confirming the strong discriminative power of neural-descriptors in large-scale environments. While Temporal Fusion (*i.e.*, sliding window) yields marginal gains, Spatial Fusion significantly increases the mIoU to 60.8%. This improvement demonstrates that our map-centric approach provides superior long-term context compared to transient temporal buffers. Notably, *UniMapping* maintains high accuracy on moving targets without explicit dynamic object removal. This suggests that our Spatial Fusion mechanism handles transient observations by leveraging the asymmetric fusion strategy, validating its role as a robust Perceptual Foundation for complex embodied tasks.

### 4.3. Sensitivity Analysis on Localization Accuracy

Spatial Fusion relies on pose estimation. We evaluate its sensitivity to localization errors by adding random noise to the poses. As shown in Figure 7 (left), detection performance degrades smoothly rather than abruptly as noise

increases. Even under a substantial perturbation of 15 cm, mAP$_{25}$ decreases by only 1.2% absolute. This indicates that the proposed approach is robust to moderate pose inaccuracies. Overall, the results suggest that SADT produces smooth feature representations, enabling Spatial Fusion to aggregate spatial features reliably and maintain stable perception performance under imperfect odometry.

### 4.4. Ablation Study

We conduct several ablation studies in which all variants are trained on 25% of the *training* set.

**Neural-Descriptor Extraction**. The quality of Neural-descriptor is fundamental to *UniMapping*. Table 5 evaluates multimodal fusion strategies on localization accuracy. Cross-attention lacks explicit spatial constraints, leading to suboptimal correspondences. While vanilla deformable attention incorporates spatial priors, it remains agnostic to perspective projection, causing sampling distortions in scenes with significant depth variance. In contrast, our SADT achieves superior accuracy by injecting explicit geometric inductive bias, thereby rectifying perspective-induced scale ambiguity and ensuring robust feature aggregation.

**Spatial Fusion Method**. Table 6 compares multiple candidates of Spatial Fusion strategies. The table displays methods for utilizing spatial information at varying degrees from top to bottom. As spatial explicitness increases, model performance improves accordingly. Notably, Transformers with limited parameters cannot explicitly leverage spatial information, whereas soft correlations degrade the semantic accuracy of descriptors. Our proposed Multi-Scale Aggregation achieves the best object detection performance.

**Runtime and Memory Analysis**. Runtime and memory usage are evaluated on a single NVIDIA GeForce RTX 3090 GPU with PyTorch framework. The feature extraction network accounts for $46.2\pm5.6$ ms. The SLAM modules introduce an overhead of $42.1\pm6.1$ ms and Spatial Fusion introduces $35.3\pm11.0$ ms. Detection and segmentation heads incur $66.2\pm4.1$ ms and $12.3\pm5.2$ ms. *UniMapping* can

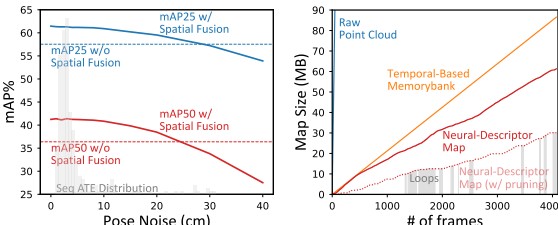

*Figure 7.* (left) **Sensitivity Analysis of Perception to Pose Noise** quantifying the resilience of neural-descriptor maps against incremental localization uncertainties, and (right) **Memory Usage** demonstrates *UniMapping* can reconstruct long-term scenes with smaller memory requirements, compared to other strategies.

achieve *approx* $5.3\,\mathrm{fps}$ inference speed with $\leq 1.4\,\mathrm{GB}$ GPU memory. As shown in Figure 7 (right), *UniMapping* minimizes overhead by storing 256 descriptors per keyframe. Keyframes comprise 11.9% of sequences, averaging *approx* $6\,\mathrm{m}$ spacing. During looping, the frequency of keyframes decreases automatically. An aggressive distance-based pruning strategy (*i.e.*, deleting keyframes within the $5\,\mathrm{m}$, *see appendix B*) can further reduce the overhead from $61\,\mathrm{MB}$ to $31\,\mathrm{MB}$, with a marginal increase in ATE by *approx* 8%.

## 5. Conclusion

We present *UniMapping*, a unified SLAM framework that bridges temporal observations into spatial perception, enabling effective localization and consistent support for downstream perception tasks.

**Limitations.** Our current map update strategy mainly relies on scan-wise descriptor replacement. A descriptor-wise fusion may better preserve historical spatial context, which is an important direction for future work. In addition, *UniMapping* currently requires separate weights for indoor and outdoor due to substantial data heterogeneity. Learning a unified weight that generalizes across diverse scene types is another promising research direction.

## Impact Statement

This work aims to advance research in SLAM and multimodal spatial representation. Such representations may benefit applications in robotics, autonomous systems, and embodied AI, and focus on fundamental representation learning. We do not foresee immediate negative societal impacts specific to this work.

## Acknowledgments

This work was supported by National Natural Science Foundation of China (No. 62576107), the Science and Technology Commission of Shanghai Municipality (No. 24511103300, No. 24511104200, and No. 25DZ2200800).

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

# A. Theoretical Analysis of Spatial-Aware Deformable Transformer (SADT)

## A.1. Derivation of Inverse-Depth Scaling

To demonstrate the physical scale invariance of the Spatial-Aware Deformable Transformer (SADT), we provide a formal derivation relating the learned offsets to metric extents via camera intrinsics.

Assume a pinhole camera model where a reference point $r$ in 3D space has coordinates $r^{\text{xyz}} = (x, y, z)$ in the camera frame. The perspective projection maps this point to image coordinates $(u, v)$ with depth $d$ as follows

$$u = f_x \frac{x}{z} + c_x, \quad v = f_y \frac{y}{z} + c_y, \quad d = z \tag{13}$$

where $(f_x, f_y)$ and $(c_x, c_y)$ denote the focal lengths and the principal point, respectively.

In standard deformable attention, the sampling offset $\Delta r^{\text{uv}} = (\Delta u, \Delta v)$ is typically learned directly from the feature $r^{\text{feat}}$. However, as shown by the differential relationship $\Delta u = f_x \frac{\Delta x}{z}$, a fixed pixel-level offset $\Delta u$ corresponds to a varying physical receptive field $\Delta x$ that scales linearly with depth $z$

$$\Delta x = \frac{z}{f_x} \Delta u \tag{14}$$

The core motivation of SADT is to enable the network to predict a depth-independent spatial offset, ensuring that the same object geometry yields consistent sampling patterns regardless of the camera-to-object distance.

*For example, consider a semantic prior where a table leg is typically located 1 meter below the tabletop, i.e., a metric displacement $\Delta y = -1m$. To capture this feature, the required pixel offset $\Delta v$ must be depth-dependent. (a) for a table at $z = 2m$, $\Delta v = -0.5 f_y$, but (b) for a table at $z = 5m$, $\Delta v$ must adaptively scale to $-0.2 f_y$. SADT allows the feature $r^{\text{feat}}$ to learn a single, unified representation of this 1-meter offset, which is then dynamically projected to the image plane.*

To achieve this, we modulate the image-plane sampling offset by the inverse depth. In SADT, the offset generation is defined as

$$\Delta r^{\text{uv}} = (\Delta u, \Delta v) = \alpha \cdot s_{\text{img}} \cdot \tanh(\text{MLP}(r^{\text{feat}})) \tag{15}$$

By setting the scaling factor $\alpha = 1/d = 1/z$, the resulting geometric offset $\Delta x$ in metric space becomes

$$\Delta x = \frac{z}{f_x} \Delta u = \frac{\cancel{z}}{f_x} \cdot \frac{1}{\cancel{z}} \cdot s_{\text{img}} \cdot \tanh(\text{MLP}(r^{\text{feat}}))_u \tag{16}$$

Similarly, the spatial offset $\Delta y$ can be derived as

$$\Delta y = \frac{s_{\text{img}}}{f_y} \cdot \tanh(\text{MLP}(r^{\text{feat}}))_v \tag{17}$$

Here, the $s_{\text{img}}$ served as a predefined hyper-parameter controlling the perception field of each query. Since $s_{\text{img}}$ and $(f_x, f_y)$ are constant for a given camera configuration, the physical receptive field $(\Delta x, \Delta y)$ is successfully decoupled from the depth $z$, thereby achieving scale invariance in feature sampling.

## A.2. Stability Discussion under Extreme Depth Values

To ensure the robustness of SADT in real-world scenarios, we analyze the stability of the inverse-depth scaling $\alpha = 1/z$ under extreme depth values.

**Case 1: Zero Depth** ($z \to 0$). When the camera is extremely close to an object, the term $1/z$ may lead to numerical instability or excessively large pixel offsets that exceed the image boundaries. However, since distortion and aberrations dominate at this stage, even correctly calculated offsets cannot extract valid features from the image. To mitigate this, we implement a depth-clipping strategy such that $\alpha = 1/\max(z, z_{\min})$, where $z_{\min}$ is typically set to the camera's near-plane distance (*i.e.*, $0.1\,\text{m}$). This ensures that the sampling remains bounded and numerically stable.

**Case 2: Infinite Depth** ($z \to \infty$). For distant background structures (*e.g.*, the horizon in KITTI), $\alpha$ approaches zero. This causes the deformable offsets $\Delta r^{\mathrm{uv}}$ to collapse toward the reference point $r^{\mathrm{uv}}$, effectively reducing the deformable attention to standard point-based sampling. This behavior is geometrically intuitive: as distance increases, the angular extent of a fixed-size physical object shrinks, and the network should naturally focus on a smaller, more localized pixel neighborhood to maintain feature saliency.

## B. Mapping Module

*UniMapping* utilizes an incremental pose-graph-based mapping strategy to ensure a consistent and reusable neural descriptor map. An online pose graph $\mathcal{G}$ is maintained, where vertices correspond to frames (keyframes or non-keyframes) and edges encode relative pose constraints. Given the current frame $F_t$, the mapping module incrementally updates $\mathcal{G}$ through four stages: (1) scan-to-keyframe odometry, (2) keyframe decision and refinement, (3) loop detection and closure, and (4, optional) pruning., as described in Algorithm 1.

**Initialization**. Neural descriptors are first extracted from the current frame $F_t$. If the pose graph $\mathcal{G}$ is empty, $F_t$ is inserted as the initial fixed keyframe with identity pose, and the algorithm terminates.

**Stage 1: Scan-to-Keyframe Odometry**. Otherwise, if $\mathcal{G}$ is non-empty, a reference keyframe is first selected for odometry. The last seen keyframe $F_{\mathrm{key}}$ is used to retrieve a local candidate set $\mathcal{F}_{\mathrm{local}}$, which is collected via a BFS with depth 5. Then, the nearest target frame $F_{\mathrm{target}}$ is selected from $\mathcal{F}_{\mathrm{local}}$. We perform scan-to-keyframe registration between $F_t$ and $F_{\mathrm{target}}$, yielding an estimated pose $T_t$ and a confidence score $\varepsilon$. In some rare cases, if $\varepsilon < \varepsilon_{\mathrm{odom}}$, the frame is discarded and the pose graph remains unchanged.

**Stage 2: Keyframe Decision and Refinement**. Otherwise, we determine whether $F_t$ should be treated as a keyframe based on its relative translation $\Delta d$ and rotation $\Delta r$ *resp.*, $F_{\mathrm{target}}$. If both $\Delta d < d_0$ and $\Delta r < r_0$, $F_t$ is treated as a non-keyframe and appended to $\mathcal{G}$ with an *odom-edge*. Otherwise, $F_t$ is considered a keyframe candidate. We collect a local map $\mathcal{F}_{\mathrm{map}}$ using BFS and perform scan-to-map registration to refine the pose estimate. If the refined confidence $\varepsilon_{\mathrm{ref}}$ improves over the initial result, the refined pose is adopted. The frame is then inserted into $\mathcal{G}$ as a keyframe with an *odom-edge*.

**Stage 3: Loop Detection and Closure**. For each new keyframe, a loop candidate set $\mathcal{F}_{\mathrm{loop}}$ is generated. Each candidate $F_{\mathrm{loop}}$ is verified via pairwise registration with $F_t$. If a candidate passes loop verification, a *loop-edge* is added to $\mathcal{G}$ and pose-graph optimization using the Levenberg-Marquardt method is applied.

**Optional Stage 4: Pruning**. When enabling key-frame pruning (*e.g.*,, as shown in Figure 7 (right)), a distance-based pruning is applied once the loop-closure is confirmed. If the distance between loop frame $F_{\mathrm{loop}}$ to current frame $F_t$ is less than a certain threshold (5 m), the $F_{\mathrm{loop}}$ vertex will be removed from pose-graph map, and all edges attached to $F_{\mathrm{loop}}$ will be disconnect and reconnected to current frame $F_t$. This procedure further reduce the memory overhead by *approx* 50%, and solve the *dynamic object* issues since outdated descriptors are removed during observing.

## C. Downstream Perception Integration Details

### C.1. V-DETR Integration for Object Detection

V-DETR (Shen et al., 2024) follows a DETR-based framework for 3D object detection, consisting of a feature encoder to extract 3D geometric representations and a Transformer decoder that iteratively refines 3D object queries through multiple layers equipped with 3D Vertex Relative Position Encoding (3DV-RPE).

We integrate the V-DETR (Shen et al., 2024) detection head (implemented by mmdetection3d (Contributors, 2020)) into our *UniMapping* framework by bypassing the original backbone and directly retrieving the multi-view fused Spatial Aggregated Descriptors from the Spatial Fusion module. To leverage the persistent map context, the Spatial Aggregated Descriptors are projected into the Transformer latent space to serve as *keys* and *values*. Crucially, we utilize the spatial coordinates of these descriptors as anchors to initialize the center predictions in the V-DETR decoder. This adaptation effectively transforms the head into a map-centric detector that utilizes SLAM-refined geometry and accumulated semantic context as priors for robust object localization.

## C.2. SPVCNN Integration for Semantic Segmentation

SPVCNN (Tang et al., 2020) adopts an architecture similar to the U-Net, comprising multiple encoder layers and multiple decoder layers with skip-connect and point-voxel interactions.

We reimplement the decoder module into our framework based on the implementation provided by mmdetection3d (Contributors, 2020). Specifically, each layer within the decoder takes (a) the features from the previous decoder layer and (b) the features from the corresponding encoder layer as input. We modified this structure by feeding multi-layer feature voxels $\mathcal{V}$ from the Geometric Encoder as input (b).

To incorporate the proposed Spatial Aggregated Descriptors, we inject them at the beginning of each decoder layer using the point-to-voxel interpolation mechanism introduced in SPVCNN.

Specifically, for each voxel center $v$, the interpolated feature $v^{\text{new}}$ is updated as a weighted aggregation of nearby Spatial Aggregated Descriptors as

$$v^{\text{new}} = \phi_3\left(v^{\text{xyz}}, \hat{\mathcal{R}}\right) + v^{\text{feat}} \tag{18}$$

Through this modification, we successfully integrated the SPVCNN semantic segmentation model into our framework and achieved positive results.

# D. Training Details

To facilitate the learning of spatial representations, we adopt a sequential spatial training strategy. Specifically, the training dataset was dynamically divided into multiple *clips*, each containing a certain spatially contiguous frames (frames within 20 meters for outdoor scenes, and frames with a field-of-view overlap ¿ 25% for indoor scenes). These clips were trained in a random order.

The model incorporates an internal cache to store past descriptors. In each batch, the descriptor of each frame is extracted and separately calculates loss with cache (*e.g.*, Pairing Loss $\mathcal{L}_{\text{p}}$ and Offset Loss $\mathcal{L}_{\text{o}}$) and without cache (*e.g.*, Loss from downstream perceptions). After calculating the total loss $\mathcal{L}$, gradients are accumulated, and these descriptors are retained in the cache for future use. Upon completing training for a clip, the cache contents are cleared, and backpropagation is performed.

We adopted gradient accumulation for two reasons: (1) We did not want descriptors inside the cache to become stale after network weight updates, and (2) Since scenes within a single clip are not randomly sampled, continuous training would violate the training's *i.i.d.* (independent and identically distributed) assumption. Gradient accumulation reduces update frequency while ensuring independence between updates, thereby minimizing this issue.

During early training, the focus is on developing the single-frame processing capability of network, thus the clip length is kept low. As training progresses and single-frame feature extraction reaches a certain level, the clip length is increased. We employed a continuous learning strategy, initially setting the clip length to 4 frames and doubling it at epochs 8 and 12, finally reaching 16 frames.

# E. Additional Experiments and Ablations

## E.1. Reference Point Generator

We investigate the impact of the Reference Point Generator (RPG) by comparing our adapted Farthest Point Sampling (FPS) strategy with different numbers $n$ against Voxel-based Downsampling with different grid sizes $s$.

First, regarding the sampling density, the results indicate a critical trade-off between spatial coverage and feature redundancy. Using a moderate number of points ($n = 256$) strikes an optimal balance. Reducing the density ($n = 128$) will expand the spatial intervals between descriptors, which often exceed the effective regression range of the Offset Head $H_{\text{offset}}$, thereby degrading registration stability. Conversely, scaling to higher densities ($n \geq 512$) leads to better performance on both SLAM and detection. However, overlapping receptive fields result in feature redundancy that incurs computational cost and storage overhead without providing more information.

Furthermore, regarding the sampling strategy, FPS demonstrates superior geometric adaptivity compared to voxel-based heuristics. While voxel-based methods provide uniform spatial partitioning, they are insensitive to the varying density of the

*Table 7.* **Ablation on Reference Point Generator**.

| | Strategy | SLAM Performance | | | | | Detection Performance[*] | |
|---|---|---|---|---|---|---|---|---|
| | | 000 | 059 | 106 | 169 | 181 | mAP$_{25}$ | mAP$_{50}$ |
| **FPS** | $n = 128$ | 13.6 | 13.0 | 9.4 | 5.0 | 4.6 | 59.56 | 39.56 |
| | $n = 256$ | 13.8 | 7.4 | 5.7 | 7.7 | 3.8 | 61.44 | 41.28 |
| | $n = 512$ | 13.8 | 7.2 | 5.5 | 8.3 | 3.5 | 61.49 | 41.38 |
| | $n = 1024$ | 13.1 | 9.1 | 5.4 | 8.5 | 3.7 | 61.31 | 41.28 |
| **Voxel** | $s = 7.5$cm | 14.5 | $\times$ | 26.1 | 9.3 | $\times$ | 60.97 | 40.26 |
| | $s = 10$cm | 13.1 | $\times$ | 4.9 | 9.1 | 3.4 | 60.70 | 39.78 |
| | $s = 15$cm | 13.3 | 8.0 | 4.3 | 10.7 | 3.5 | 59.91 | 38.95 |

[*] To isolate the pose estimation errors introduced by SLAM, we directly utilize ground-truth poses here to independently measure the impact of reference point generators.

underlying manifold, resulting in unstable localization performance across diverse scenes. In contrast, FPS ensures that the descriptors are maximally distributed across the point cloud manifold. This manifold-constrained sampling provides a robust and consistent spatial support for both the localization and downstream detection heads.

### E.2. Detailed Results of Object Detection

Table 8 presents detailed 3D object detection results on the ScanNet benchmark, a large-scale indoor RGB-D dataset. Based on the raw video sequences and detection labels provided by the benchmark, we projected scene bbox labels onto each frame of the observation and trained the detection model as well as baseline V-DETR (Shen et al., 2024).

*Table 8.* **Accuracy of Detection Task on ScanNet Benchmark (AP↑)**.

| Method | # param. | mAP25 | mAP50 | picture | curtain | s.curtain | sink | toilet | chair | cabinet | counter | desk | table | door | window | sofa | bed | bk.shelf | bathtub | fridge | oth.furn. |
|---|---|---|---|---|---|---|---|---|---|---|---|---|---|---|---|---|---|---|---|---|---|
| V-DETR | 75.44M | 57.4 | 39.1 | 32.4 | 41.2 | 55.6 | 65.3 | 98.4 | 84.7 | 42.3 | 34.7 | 64.3 | 55.2 | 51.4 | 32.9 | 82.2 | 78.0 | 38.7 | 86.9 | 42.6 | 45.5 |
| ours (w/o SF) | 61.26M | 57.7 | 36.6 | 35.8 | 49.3 | 44.3 | 65.4 | 97.7 | 82.3 | 46.5 | 34.6 | 61.8 | 52.9 | 49.9 | 37.9 | 79.2 | 76.0 | 44.9 | 86.5 | 48.2 | 46.2 |
| ours (w/ TF) | 62.39M | 59.2 | 39.0 | 38.1 | 49.8 | 46.5 | 68.4 | 97.8 | 83.0 | 48.6 | 36.5 | 65.6 | 54.9 | 52.1 | 39.5 | 81.1 | 76.7 | 44.8 | 86.3 | 49.8 | 47.0 |
| ours (w/ SF) | 62.39M | 60.8 | 40.4 | 38.6 | 51.4 | 49.0 | 71.0 | 97.7 | 83.7 | 49.1 | 40.4 | 67.8 | 56.3 | 52.6 | 41.6 | 82.0 | 78.0 | 46.2 | 87.4 | 53.2 | 48.2 |

We compare the single-frame baseline V-DETR with two variants of our method, *i.e.*, with and without the proposed Spatial Fusion (SF) module. While using fewer parameters than V-DETR. The variant without Spatial Fusion performs comparably to the baseline, indicating that the unified architecture alone already provides competitive detection capability. Incorporating Spatial Fusion further yields consistent gains across most object categories, particularly for small or cluttered indoor objects such as curtain, counter, and bookshelf, demonstrating the effectiveness of aggregating spatially accumulated multi-view features for dense indoor detection tasks. Overall, our full model with Spatial Fusion achieves the best performance, improving both mAP25 and mAP50.

### E.3. Detailed Results of Semantic Segmentation

Table 9 reports detailed semantic segmentation results on the SemanticKITTI benchmark, a large-scale outdoor dataset collected from LiDAR sensors in urban driving scenarios.

It is worth noting that since we need to jointly train SLAM as well as segmentation task, we adopted the SLAM data split (*i.e.*, `00-05` for training, and `06-10` for evalation) rather than the official splitting (*i.e.*, `00-10` for training, and others for evaluation).

The SPVCNN implemented by mmdetection (Contributors, 2020) was retrained as a baseline. The proposed approach consistently outperforms this baseline across most categories, achieving a substantial improvement in mean IoU. Incorporating Spatial Fusion results in a clear performance gain over the variant without fusion, particularly for small or sparsely observed objects such as *person*, *bicyclist*, and *pole*. This finding demonstrates the benefit of aggregating multi-frame spatial context.

We observe that the *motorclist* and *other-ground* class exhibits *extremely* low IoU across all methods. This is primarily

*Table 9.* **Accuracy of Semantic Segmentation Task on SemanticKITTI Benchmark (IoU↑).**

| Method | mIoU | car | bicycle | motorcycle | truck | other-vehicle | person | bicyclist | motorcyclist | road | parking | sidewalk | other-ground | building | fence | vegetation | trunk | terrain | pole | traffic-sign |
|---|---|---|---|---|---|---|---|---|---|---|---|---|---|---|---|---|---|---|---|---|
| SPVCNN | 50.7 | 91.0 | 11.0 | 55.8 | 3.2 | 28.4 | 40.7 | 41.3 | 0.0 | 86.9 | 53.3 | 76.1 | 0.2 | 89.2 | 68.2 | 81.3 | 61.0 | 67.9 | 62.8 | 44.3 |
| ours (w/o SF) | 53.7 | 89.5 | 15.8 | 60.9 | 47.7 | 46.3 | 44.6 | 65.2 | 0.0 | 85.8 | 39.1 | 71.5 | 5.2 | 83.1 | 51.8 | 80.0 | 63.8 | 70.7 | 63.2 | 36.2 |
| ours (w/ TF) | 53.8 | 89.6 | 15.3 | 60.6 | 48.4 | 46.7 | 44.5 | 65.9 | 0.0 | 85.8 | 39.0 | 71.5 | 5.3 | 83.0 | 51.7 | 79.9 | 63.8 | 70.7 | 63.2 | 36.5 |
| ours (w/ SF) | 60.3 | 90.2 | 16.7 | 63.7 | 54.3 | 56.1 | 64.2 | 76.1 | 0.2 | 89.2 | 51.9 | 77.3 | 7.3 | 86.1 | 55.3 | 81.8 | 69.4 | 76.0 | 70.1 | 59.2 |

due to its low occurrence frequency ($< 0.005\%$) in our used training set (*i.e.*, `00-05`). Compared to models trained on the official SemanticKITTI split, which provides broader exposure to this class, our model receives limited supervision for motorcyclists, resulting in lower segmentation performance.

The visual results are presented in Figure 8. Benefiting from our Spatial Fusion strategy, *UniMapping* effectively maintains spatial consistency even in the presence of dynamic entities. These moving objects are correctly recognized and integrated without introducing any ghosting artifacts or blurring, which are common failure cases in traditional feature accumulation.

## F. Notations and Hyperparameter

Finally, we provide the notations and hyperparameter settings used in our paper in Table 10.

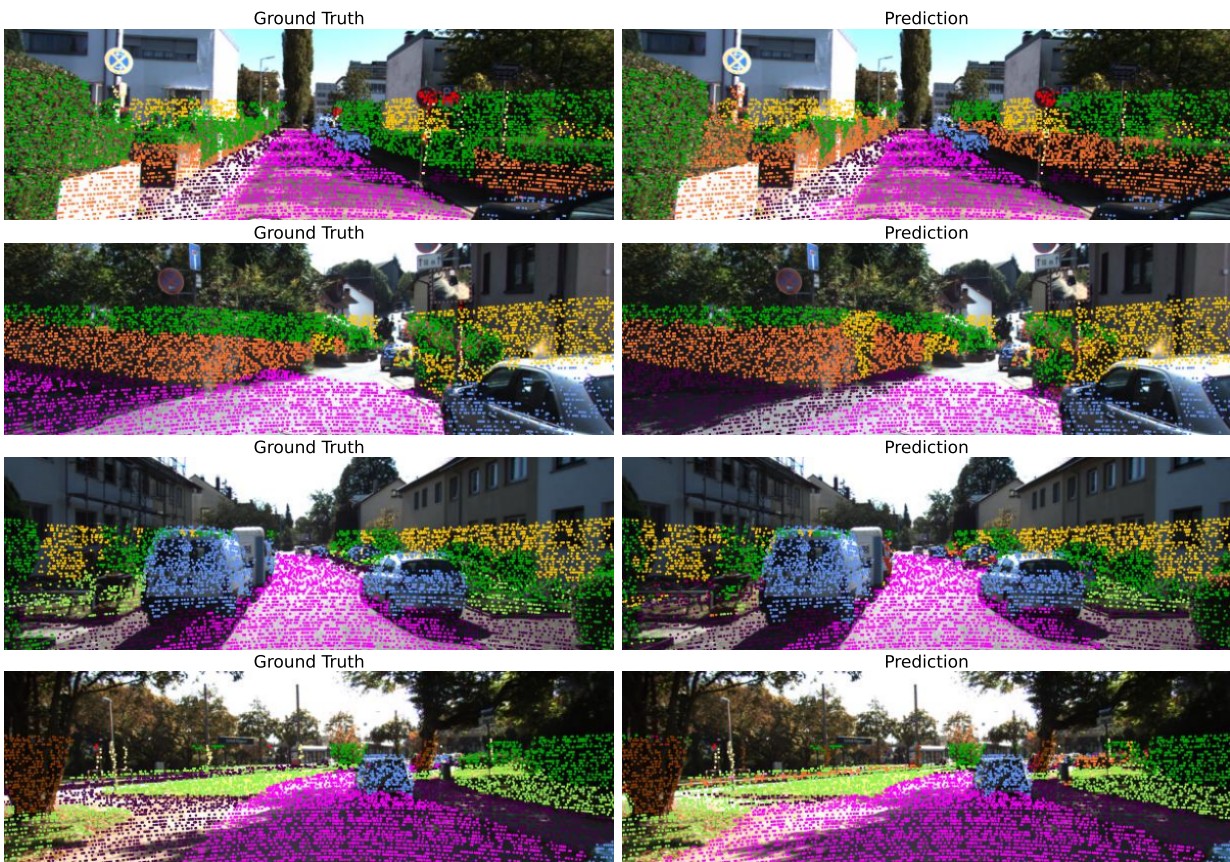

*Figure 8.* Visualization of Semantic Segmentation.

*Table 10.* **Summary of Notations and Hyperparameters**.

| Symbol | Description | Value |
|---|---|---|
| **Basic** | | |
| $\mathcal{R}$ | Neural-Descriptors | |
| $r^{\mathrm{xyz}}$, $r^{\mathrm{uvd}}$ and $r^{\mathrm{feat}}$ | 3D position / feature of Neural-Descriptor | |
| $r^{\mathrm{img}}$ and $r^{\mathrm{geo}}$ | Aggregated feature from image and pointcloud | |
| **Feature Extraction** | | |
| $\mathcal{I}$ | Image | |
| $\mathcal{P}$ | Point Cloud | |
| $\mathcal{X}_i$ | Feature map derived from Image | |
| $x^{uv}$ | 2D position of feature pixel | |
| $\mathcal{V}_i$ | Feature voxel derived from pointcloud | |
| $v^{xyz}$ | 3D position of feature voxel | |
| $\mathcal{R}_0$ | Reference points | |
| $r_0^{\mathrm{feat}}$ and $r_0^{\mathrm{xyz}}$ | Feature / Position of reference point | |
| $[R_{\mathrm{cam}}|T_{\mathrm{cam}}]$ | Camera extrinsics | |
| $K_{\mathrm{cam}}$ | Camera intrinsics | |
| $\pi(\cdot)$ | denotes perspective division | |
| $\alpha$ | Perspective-aware scaling factor | |
| $s_{\mathrm{img}}$ | Scale factor for image offsets | 8.0 |
| $\phi_2(\cdot)$ | Bilinear sample in image feature map | |
| $N_{\mathrm{offset}}^{\mathrm{img}}$ | Number of image offsets | 4 |
| $A_{i,h,k}^{\mathrm{img}}$ | Attention weight for image feature | |
| $s_{\mathrm{geo}}$ | Scale factor for geometric offsets | 2.0 |
| $N_{\mathrm{offset}}^{\mathrm{geo}}$ | Number of geometric offsets | 8 |
| $\phi_3(\cdot)$ | Trilinear sample in geometric feature voxel | |
| $A_{i,h,k}^{\mathrm{geo}}$ | Attention weight for geometric feature | |
| **Mapping** | | |
| $\mathrm{H}_{\mathrm{pair}}$ | Pairing network | |
| $\sigma_{ij}$ | Confidence score between descriptor $i$ and map point $j$ | |
| $\varepsilon$ | Confidence of the registration between two frames | |
| $\varepsilon_{\mathrm{odom}}$ | Odometry confidence threshold | 0.6 |
| $\varepsilon_{\mathrm{loop}}$ | Loop-closure confidence threshold | 0.75 |
| $\mathrm{H}_{\mathrm{offset}}$ | Offset network | |
| $\delta_{ij}$ | Predicted offset between descriptor $i$ and map point $j$ | |
| $R, T$ | Estimated rotation and translation between two frames | |
| $\omega_{ij}$ | Weight for solving relative pose between frame $i$ and $j$ | |
| $\sigma$ | Robust scale parameter for Geman-McClure loss | |
| **Perception Tasks** | | |
| $\hat{\mathcal{R}}$ | Spatial Aggregated Descriptors | |
| $\mathcal{Y}$ | Result of perception task | |
| $d_0$ | Euclidean range of Spatial Fusion | 0.25m, 0.50m, 1.00m for ScanNet 5m, 10m, 20m for SemanticKITTI |
| **Losses** | | |
| $\tau$ | Temperature parameter for contrastive loss | 0.1 |
| $d_0$ | Positive distance threshold | 7.5cm for ScanNet and 2m for SemanticKITTI |
| $\mathcal{L}_c$ and $\mathcal{L}_o$ | Contrastive loss and Offset loss | |
| **Operators** | | |
| $\oplus$ | Concatenation | |
| $\odot$ | Cosine similarity | |

---

**Algorithm 1** Online Mapping and Pose Graph Maintenance

---

**Require:** Current frame $F_t$, Pose Graph $\mathcal{G}$
**Ensure:** Updated Pose Graph $\mathcal{G}$, Current Pose $T_t$
 1: **Initialize:** Extract neural descriptors of $X_t$ via Feature Extraction Network.
 2: **if** $\mathcal{G}$ is empty **then**
 3:     $T_t \leftarrow I$, add $F_t$ as the first fixed keyframe to $\mathcal{G} = \{F_t\}$.
 4:     **return** $T_t, \mathcal{G}$
 5: **end if**
 6: {*Stage 1: Scan-to-Keyframe Odometry*}
 7: Get the last seen keyframe $F_{\text{key}} \in \mathcal{G}$
 8: $\mathcal{F}_{local} \leftarrow \texttt{BFS\_Search}(\mathcal{G}, F_{\text{key}}, \text{depth} = 5)$
 9: $F_{target} \leftarrow \texttt{Select\_Candidate}(\mathcal{F}_{local})$
10: $T_t, \varepsilon \leftarrow \texttt{Registration}(\{F_t\}, \{F_{target}\})$
11: **if** $\varepsilon < \varepsilon_{\text{odom}}$ **then**
12:     **return** $T_t, \mathcal{G}$
13: **end if**
14: {*Stage 2: Keyframe Decision and Refinement*}
15: $\Delta d \leftarrow \texttt{Distance}(F_t, F_{target})$
16: $\Delta r \leftarrow \texttt{Rotation}(F_t, F_{target})$
17: **if** $\Delta d < d_0$ **and** $\Delta r < r_0$ **then**
18:     Append non-keyframe vertex $F_t$ and *odom-edge* $(F_t, F_{target}, T_t)$ into $\mathcal{G}$
19:     **return** $T_t, \mathcal{G}$
20: **end if**
21: $\mathcal{F}_{map} \leftarrow \texttt{BFS\_Search}(\mathcal{G}, F_t, \text{depth} = 5)$
22: $T_{\text{ref}}, \varepsilon_{\text{ref}} \leftarrow \texttt{Registration}(\{F_t\}, \mathcal{F}_{map})$
23: **if** $\varepsilon_{\text{ref}} < \varepsilon$ **then**
24:     $T_t \leftarrow T_{\text{ref}}$
25: **end if**
26: Append keyframe vertex $F_t$ and *odom-edge* $(F_t, F_{target}, T_t)$ into $\mathcal{G}$
27: {*Stage 3: Loop Detection and Closure*}
28: $\mathcal{F}_{loop} \leftarrow \texttt{Loop\_Candidate\_Search}(F_t)$
29: **for all** $F_{loop} \in \mathcal{F}_{loop}$ **do**
30:     $T_{loop}, \varepsilon_{\text{loop}} \leftarrow \texttt{Registration}(F_t, F_{loop})$
31:     **if** $\texttt{Verify\_Loop}(F_t, F_{loop}, \varepsilon_{\text{loop}})$ **then**
32:         Append *loop-edge* $(F_t, F_{loop}, T_{loop})$ into $\mathcal{G}$
33:         Apply Pose-Graph Optimization on $\mathcal{G}$
34:         **break**
35:     **end if**
36: **end for**
37: **return** $T_t, \mathcal{G}$

---

