# OpenReview forum: "UniMapping: Unified SLAM Framework for Map-Centric Embodied Perception"
_ICML.cc/2026/Conference — ICML 2026 spotlight_

### Official Review · Reviewer_k335 · 2026-02-24

**Soundness:** 3
**Presentation:** 2
**Significance:** 2
**Originality:** 2
**Overall Recommendation:** 6
**Confidence:** 3

**Summary:**

UniMapping presents a highly motivated and conceptually elegant framework that unifies SLAM and downstream semantic perception through a shared neural-descriptor map, its practical execution relies on ad-hoc heuristics that compromise its methodological rigor. Although the transition from temporal to spatial fusion via the Spatial-Aware Deformable Transformer (SADT) is physically intuitive, the framework exhibits critical vulnerabilities: the SADT formulation risks losing actual scale-invariance during cross-camera generalization; the registration mechanism utilizes statistically fragile confidence metrics lacking proper geometric uncertainty modeling; the Spatial Fusion module's Max-Pooling operation is theoretically prone to semantic feature smearing from lingering dynamic objects; and “the naive 5-meter” trajectory-based map pruning is an unsound logical leap for dynamic scene understanding.

The authors must explicitly address these specific methodological flaws with robust ablation studies or theoretical justifications during the rebuttal, and add the "Limitations" section to transparently reflect the framework's boundaries in handling non-loop dynamics and unconstrained, dynamic environments.

**Compliance With Llm Reviewing Policy:**

Affirmed.

**Final Justification:**

gogogo

**Key Questions For Authors:**

1.How does the SADT framework maintain actual scale-invariance when deployed on cameras with different focal lengths ($f_B \neq f_A$)? Please clarify how the model avoids unintentionally scaling the physical receptive field when the training camera's focal length is implicitly absorbed by the MLP weights.

2.How does the system prevent false-positive loop closures in environments with repetitive structures where the "average similarity" confidence metric ($\epsilon$) is artificially high but geometric constraints are ambiguous? Additionally, how does the MLP-based 3D offset prediction ($\delta_{ij}$) generalize to unseen metric scales without explicit geometric constraints?

3.Could you justify the arbitrary “5-meter threshold” for map pruning with ablation studies across fundamentally different spatial scales (e.g., ScanNet vs. SemanticKITTI)? Furthermore, how does the framework eliminate "ghost descriptors" of dynamic objects in exploratory trajectories where loop closures are never triggered?

4.How does the Spatial Fusion module prevent semantic "feature smearing" when a static background query (e.g., the ground) retrieves lingering dynamic "ghost" features from the historical map?  Please explain how the framework prevents these high-intensity ghost semantics from overriding background features during the Max-Pooling operation.

5.Does the proposed UniMapping framework inherently overfit to the geometric layouts of its training datasets? Please provide quantitative evidence or zero-shot evaluation results demonstrating that the learned neural descriptors and the offset prediction network ($H_{offset}$) can robustly generalize to entirely unseen environments (e.g., cross-city or cross-dataset evaluations) without any scene-specific fine-tuning or retraining.

**Limitations:**

Please refer to the "weakness".

**Strengths And Weaknesses:**

**Strengths:**

This paper presents UniMapping, a highly forward-looking and physically grounded map-centric unified framework in the field of Embodied Perception.  Its core strength lies in its attempt to bridge the long-standing gap between traditional SLAM mapping and downstream semantic perception tasks. It proposes a unified Neural-Descriptor Map that simultaneously serves pose estimation and downstream tasks such as 3D object detection and segmentation.

The authors innovatively introduce a Spatial-Aware Deformable Transformer (SADT), which incorporates geometric inductive biases (e.g., inverse-depth scaling) to explore scale-invariant feature extraction. More importantly, the proposed Spatial Fusion strategy elegantly shifts the aggregation of multi-frame contexts from the unbounded temporal domain to the structured spatial domain. This design is theoretically compelling, as it effectively circumvents the spatial misalignment and computational redundancy issues inherently associated with pure Temporal Fusion in long-sequence and loop-closure scenarios.

Overall, the paper is well-motivated, offering a inspiring architectural perspective for addressing cross-task feature reuse and spatial consistency in embodied AI systems.

**Weakness:**

**Comment 1: “Incomplete Decoupling of Perspective Distortion and Cross-Camera Generalization in SADT”**

The proposed Spatial-Aware Deformable Transformer (SADT) introduces an inverse-depth scaling factor ($\alpha = 1/z$) to neutralize perspective distortion, ensuring a scale-invariant physical receptive field. However, there is a critical limitation in the current mathematical formulation regarding cross-camera generalization.As derived in Appendix A.1, the actual physical offset is $\Delta x = \frac{s_{img}}{f_x} \cdot \tanh(MLP(r^{feat}))_u$. The authors argue that because $f_x$ is a constant for a given camera configuration, the physical receptive field is decoupled from depth. This assumption forces the MLP weights to implicitly absorb the specific focal length ($f_x, f_y$) of the training camera.Consequently, if a model trained on Camera A ($f_A$) is deployed on Camera B ($f_B$) with a different focal length, the actual physical receptive field will be unintentionally scaled by a factor of $(f_A / f_B)$. This breaks the scale-invariance property when transferring across different sensor setups, severely limiting the framework's claim as a 'generalizable foundation' for embodied perception.

**Comment 2: “Flawed Registration Mechanism and Confidence Metric”**

The scan-to-scan registration heavily relies on ad-hoc heuristics rather than principled geometric or probabilistic constraints. Specifically, predicting metric-space 3D offsets $\delta_{ij}$ directly via an MLP ($H_{offset}$) from concatenated features is a black-box approach that lacks physical grounding and is highly prone to overfitting to training domain scales. Furthermore, defining the global registration confidence $\epsilon$ merely as the "average similarity over all pairs" is statistically fragile. This metric fails to model true correspondence uncertainty, making the system highly vulnerable to catastrophic false-positive loop closures in environments with repetitive structures, where average feature similarity is artificially high but geometric constraints are completely ambiguous.

**Comment 3: “Unsound Handling of Dynamic Objects”**

The authors claim in Appendix B that an aggressive distance-based pruning strategy—deleting loop keyframes within a 5-meter threshold —can inherently "solve the dynamic object issues". This is a severe logical leap that confuses simplistic memory management with principled dynamic scene understanding. This ad-hoc heuristic introduces inevitable vulnerabilities.

Arbitrary Thresholds and Lack of Ablation: The 5-meter threshold is an ad-hoc "magic number". It is highly unreasonable to apply the exact same distance metric across fundamentally different spatial scales, such as confined indoor rooms (ScanNet) and vast outdoor driving scenarios (SemanticKITTI). Furthermore, the paper provides absolutely no ablation studies to justify the optimality or robustness of this specific parameter.

Destruction of Static Landmarks (False Positives): By blindly deleting historical loop vertices, the system recklessly discards valuable static geometric details and structural landmarks just to clear potential "ghosts". This prevents the system from refining its spatial memory through repeated multi-view observations.

Failure on Non-Loop Dynamics (False Negatives): The pruning mechanism strictly requires a loop closure to trigger. It fundamentally fails for dynamic objects in exploratory trajectories. For example, if a robot is driving forward on a long straight road and is overtaken by a fast-moving car, the distance-based pruning will never be triggered. The "ghost descriptors" of the moving car will permanently remain in the global map, continuously interfering with downstream perception tasks.

**Comment 4: “Potential Feature Smearing in Spatial Fusion and Handling of Ghosting Artifacts”**

The authors propose an "asymmetric fusion" mechanism to mitigate ghosting artifacts. Restricting query anchors exclusively to current physical observations is an intuitive and partially effective strategy. It cleverly avoids generating queries in the empty spaces left by departed dynamic objects, which successfully bypasses a portion of the floating "ghost" features.
However, there are potential theoretical limitations regarding the Max-Pooling operation during feature aggregation that warrant further discussion.  In dynamic scenarios, when a moving object (e.g., a car) departs, the current sensor observes the newly exposed physical background (e.g., the ground). If this ground query performs a ball search, it might still retrieve the lingering "ghost car" descriptors from the global map due to spatial proximity.
Since the Max-Pooling operation inherently tends to retain the strongest semantic activations within its receptive field, there is a risk that the high-intensity semantics of the "ghost car" might override the background features. This could lead to semantic feature smearing, where the downstream perception head receives a ground point imbued with "car" semantics. This theoretical risk suggests that asymmetric fusion alone might not fully resolve the ghosting issue in dense scenes, which also helps explain why the distance-based pruning strategy in Appendix B serves as a necessary supplement for the system's overall robustness.

---

> ### Author Rebuttal · Authors · 2026-03-28
>
> We thank k335 for the rigorous and insightful analysis. We have addressed all concerns with (1) Mathematical analysis for scale-invariance (W1,Q1), (2) Zero-shot eval for generalization (Q5), (3) Experiments of registration (W2,Q2); and (4) Ablations on dynamics (W3-4,Q3-4). We believe these responses provide a more principled foundation for our claims.
> # R1: Decoupling of Perspective & Cross-Camera Generalization (W1, Q1)
> The MLP implicitly absorb $f$ during training. We want to clarify that:
> - **Intra-frame (Dynamic)**: The design objective of SADT is to achieve Intra-frame scale-invariance, decoupling the pixel-wise depth ($d$) and physical receptive fields.
> - **Inter-sensor (Static)**: This dependency is strictly linear and multiplicative, not entangled with the learned non-linear mapping. Therefore, it can be *exactly* reparameterized across unseen cameras via a deterministic scaling: $\alpha_\text{new} = \frac{f_\text{ref}}{f_\text{new}} \cdot \alpha_\mathrm{old}$.
>
> We conduct a cross-sensor shift on ScanNet by artificially scaling the focal length ($f_{new} = 1.5 f_\text{ref}$) and adjusting cam FoV accordingly.
>
> |Method|ATE|mAP25|mAP50|
> |-|-|-|-|
> |SADT ($f_\text{ref}$)|7.40|60.8|40.4|
> |SADT ($f_\text{new}$)|7.64|60.9|40.9|
> |SADT w/ decoupling ($f_\text{new}$)|7.40|61.0|41.0|
>
> Without decoupling, the model showed resilience due to the generalization capacity of the neural network, with a minor performance drop without retraining. Explicit decoupling perfectly restores scale-invariance. Another zero-shot exp with a different camera $f$ can be found in R5
>
> We will formally include this $\alpha_{new}$ reparameterization in the final manuscript, resolving the cross-camera generalization limitation.
> # R2: Registration, Confidence and Loop (W2, Q2)
> Our registration is a geometrically-constrained optimization, not a black box. The MLP ($H_\text{offset}$) predicts local residual corrections, and the final pose is solved via a Robust Weighted SVD. On SemanticKITTI, **offset head reduces the raw average residual between corresponding points from 57.6cm to 14.3cm (↓75.1%)**, acting as a learned prior before geometric solving.
>
> We apologize that the manuscript did not clearly present the full registration pipeline. The system does not rely on average similarity alone. It follows (1) Semantic similarity threshold, and (2) Rigid Geometric Consistency (inlier RMSE<0.25m/7.5cm after SVD). The loop is rejected if it violates rigid geometric constraints, yielding a **Loop Closure AUC of 0.97**.
> # R3: Pruning Strategy (W3, Q3)
> We clarify that pruning is not our primary approach for handling dynamics, but an optional optimization for memory efficiency in unbounded outdoor scenes. Dynamic suppression can be found in R4.
>
> The 5m threshold is not a "magic number" but a statistically grounded in the keyframe distribution. On SemanticKITTI, keyframe selection yields an avg keyframe spacing of ~6m (selection protocol was rigorously ablated for accuracy-memory tradeoffs in *DeepPointMap (supp)*). The pruning threshold must be less than this spacing to maintain the vertex density while ensuring structural continuity. That's why we use 5m to intentionally leave a 1m redundancy margin and safely discard redundant historical vertices.
>
> Furthermore, for indoor ScanNet, spatial proximity does not imply redundancy due to limited view frustums. This distance-based pruning is disabled indoors.
> # R4: Dynamic Objects (W4, Q4)
> UniMapping’s end-to-end training ensures descriptors are semantic-aware. To verify implicit ghost handling, we compared our system against an explicit "dynamic label filter". The similar performance proves the fusion mechanism inherently suppresses ghosts, making manual pruning unnecessary for accuracy.
>
> |Method|mIoU|car|bicyclist|person|
> |-|-|-|-|-|
> |SPVCNN|50.7|91.0|41.3|40.7|
> |ours|60.3|90.2|76.1|64.2|
> |ours w/ filter|60.3|90.1|75.2|64.1|
>
> To address the concern about max-pooling, we evaluated mean-pooling (less sensitive to peak intensities). The performance gap is negligible, which stability indicates that the robustness stems from the asymmetric query-anchor design and descriptor discriminability rather than a specific pooling heuristic.
>
> |Method|mAP25|mAP50|mAR25|mAR50|
> |-|-|-|-|-|
> |MaxPooling|48.8|30.6|69.1|45.9|
> |MeanPooling|48.4|30.7|68.3|45.9|
>
> # R5: Generalization (Q5)
> To address the concern about overfitting, we conduct a zero-shot eval on the KITTI-CARLA dataset (different scene & cam$f$). UniMapping trained on SemanticKITTI is directly evaluated on unseen CARLA Sequences Town0x (Tx) without any fine-tuning/scene-specific adaptation.
>
> |Metric|T1|T2|T3|T4|T5|T6|
> |-|-|-|-|-|-|-|
> |ErrR (deg)|1.5|2.7|3.5|3.9|1.1|2.7|
> |ErrT (m)|0.7|1.5|1.9|1.1|0.8|0.8|
> |RTE (cm)|5.0|5.5|5.1|9.6|6.5|6.7|
>
> Despite real-to-sim shift and varying intrinsics, UniMapping **maintained mean RTE of 6.4cm**, suggesting the model learns geometry-aware and transferable spatial representations, rather than overfitting to dataset layouts.

---

> > ### Author Rebuttal · Reviewer_k335 · 2026-04-01
> >
> > Most questions have been sloved. And you should provide a visualization for ghost handling. The content of your rebuttal should add to your appendix, and code should be opensource.

---

> > > ### Author Response · Authors · 2026-04-01
> > >
> > > Thank you very much for reviewer k335 thorough review and for updating your assessment. We are glad that our responses and additional experiments successfully addressed your concerns. Regarding your follow-up suggestions, we completely agree and commit to the following:
> > >
> > > # R1: Visualization of Ghost Handling (Q1)
> > > Thank you for this excellent suggestion. We have generated a visualization demonstrating UniMapping's performance on dynamic objects, which can be viewed via this [anonymous link](https://anonymous.4open.science/r/AnonymousRebuttal_26ICML_UniMapping-BD22/Fig%20r1%20Handling%20Dynamics.pdf).
> > >
> > > We will add a section in the Appendix to demostrate this. This clearly illustrates how UniMapping seamlessly suppresses ghost artifacts from moving objects (e.g., departing vehicles) while maintaining the integrity of the newly exposed static background.
> > >
> > > # R2: Appendix Expansion and Manuscript Revision (Q2)
> > > The registration details, the detailed pruning strategy, the handling of dynamics, and all zero-shot evaluation results (KITTI-CARLA) from the previous responses will be fully integrated into the final Appendix. Furthermore, the mathematical derivations for cross-camera reparameterization will be directly appended to the main manuscript. We will also formally recognize the reviewers' constructive feedback in the Acknowledgments section.
> > >
> > > # R3: Open Source Code (Q3)
> > > We affirm our commitment to open-source our UniMapping codebase upon publication.
> > >
> > > *Thank you again for helping us significantly strengthen the methodological rigor and transparency of our paper. Should you have any remaining questions, please feel free to let us know.*

---

### Official Review · Reviewer_Qzze · 2026-03-08

**Soundness:** 3
**Presentation:** 2
**Significance:** 2
**Originality:** 3
**Overall Recommendation:** 5
**Confidence:** 3

**Summary:**

This paper proposed a unified model which integrates SLAM and perception systems together. A spatial-aware deformable transformer is designed to extract scale-invariant neural descriptors and a spatial fusion module combines multi-frame temporal cues. Authors prove the effectiveness of proposed method on both outdoor and indoor scenarios.

**Compliance With Llm Reviewing Policy:**

Affirmed.

**Final Justification:**

Thanks for the rebuttal authors provided. I believe it is a great work facilitating both SLAM and perception domains. Although in terms of localization metrices, I still hope authors could add comparisons with more latest methods like DVLO (ECCV'24), DVLO4D (ICRA'25), the overall experiments and universality are enough to convince me. Thus, I raise my score for accept.

Also, I agree with another reviewer that code should be open-sourced to facilitate downstream tasks.

**Key Questions For Authors:**

As above, more explanations about a unified model, additional comparisons with recent SOTA localization methods, and universal provement would be my main concern.

**Limitations:**

Authors didn't provide any limitation claims in the main script.

**Strengths And Weaknesses:**

Strengths:

1. The combination of SLAM systems and percetion task in a unified model seem to be novel in this field.

2. The map construction of neural descriptors can maintain scale-invariant features.

3. Experiments are comprehensive and Table 3 & 4 demonstrates its effectiveness on detection and segmentation fields.

Weakness:

1. I am still a littl confused the significance of combining SLAM and perception tasks. Compared to decoupled methods which separately achieve both tasks, what are advantages in a unified model?

2. The compared methods are out-of-date in localization benchmark. More advanced models should be compared.

3. For me, the map reconstruction based on neural descriptors is similar to existing descriptor-based methods in SLAM (e.g., some relocalization methods which extract a descriptor like using NetVLAD to represent the scenes. What are the differences?

4. More perception metrics on different baselines are expected. Table 3 &4 only show one baseline respectively for each task, which is not so convincing about the universal effectiveness on various baeline mathods.

---

> ### Author Rebuttal · Authors · 2026-03-31
>
> We sincerely thank Reviewer Qzze for acknowledging the value of our work and providing constructive feedback. Your questions regarding the core advantages of the unified model and task generalizability are central to our contributions.
>
> # R1: Unified Model (W1)
> Compared to decoupled pipelines, UniMapping provides three key advantages in architecture:
> - **Computational Efficiency**. Decoupled systems extract features twice (once for SLAM, once for perception). UniMapping extracts once into a shared persistent map, significantly reducing redundant backbone overhead.
> - **Spatial Memory**. Single-frame models struggle with occlusions. UniMapping’s map-centric representation acts as long-term spatial memory, allowing perception heads to leverage historical context to "see" through current occlusions. (manuscript Fig. 6)
> - **Unified Geometry-Semantic Consistency**. By jointly training SLAM and perception, our shared descriptors simultaneously encode robust geometric structures and rich semantic information within a single representation.
>
> As demonstrated in Table 3 & 4, and an additional table in R4, **combining UniMapping to existing perception method (SPVCNN, VDETR and PTv3) significantly improve the performance (+10.1% mIoU & +3.4% mAP & +2.3% mIoU)**. UniMapping replaces frame-centric processing with a spatially persistent representation.
>
> # R2: Updated Localization Benchmark (W2)
> We agree that comparing against the latest SOTA is crucial. We update our outdoor evaluation (on SemanticKITTI) to include recent 2023-2025 top-tier methods.
> |Method|mean Trans|06.T|06.R|07.T|07.R|08.T|08.R|09.T|09.R|10.T|10.R|
> |-|-|-|-|-|-|-|-|-|-|-|-|
> |DELO (2023)|1.18|0.83|0.35|0.58|0.41|1.36|0.64|1.23|0.57|1.53|0.90|
> |NALO-VOM (2023)|1.09|1.33|-|1.59|-|0.90|-|1.02|-|0.85|-|
> |TransLO (2023)|0.99|-|-|0.55|0.43|1.29|0.50|0.95|0.46|1.18|0.61|
> |DiffLO (2025)|0.71|-|-|**0.37**|**0.27**|1.12|0.44|0.68|0.28|**0.66**|**0.32**|
> |UniMapping (ours)|**0.67**|**0.44**|**0.20**|0.38|0.28|**0.73**|**0.42**|**0.66**|**0.22**|0.91|0.40|
>
> *(Note: .T=Trans .R=Rot. '-' denotes metrics not reported in the respective original papers / used as part of their training set)*
>
> We also update the performance table for indoor scenes (ScanNet dataset, ATE rmse). UniMapping achieves the lowest mean ATE errors across recent SOTA methods.
> |Method|mean ATE|00|59|06|169|181|
> |-|-|-|-|-|-|-|
> |**Implict Methods**|
> |MIPS-Fusion (2023)|10.4|7.9|10.7|9.7|9.7|14.2|
> |Point-SLAM (2023)|12.7|10.2|7.8|8.7|22.2|14.8|
> |**Explicit Methods**|
> |MonoGS (2024)|16.7|9.8|32.1|8.9|10.7|21.8|
> |LoopSplat (2025)|8.0|**6.2**|**7.1**|7.4|10.6|8.5|
> |UniMapping (ours)|**7.6**|13.7|7.4|**5.7**|**7.7**|**3.7**|
>
> While the baselines are specialized for either outdoor or indoor, UniMapping is a cross-modal framework generalizing across both scenarios via a single architecture, simultaneously providing a dense perception foundation.
>
> # R3: Descriptor-Based SLAMs (W3)
> UniMapping differs from existing descriptor-based SLAM or relocalization methods in the following aspects:
> - **Local Descriptors**. Instead of global descriptors (e.g., NetVLAD), UniMapping maintains spatially indexed local descriptors anchored in 3D space, providing geometric and semantic granularity required for dense reconstruction and point-level perception.
> - **Semantic-Aware**. The representation directly supports dense perception tasks (e.g., det. and seg.), rather than post-hoc retrieval.
> - **Jointly-Optimized**. Our descriptors are not only used for matching, but are continuously fused and optimized as part of the mapping process, forming a dense neural representation of the environment.
>
> # R4: Universal Effectiveness & Additional Baselines (W4)
> To prove that our performance gains are a universal benefit (not only to the chosen baseline), we deliberately integrated UniMapping into three distinctly different experiments, spanning across environments (In/Outdoor), task dimensionalities (High-/Low-level Bbox/Dense), and backbone paradigms (Attention / Convolution):
>
> - V-DETR (Indoor, High-level Det., Attn.-based): **+3.4% mAP** manuscript (Table 3).
>
> - SPVCNN (Outdoor, Low-level Seg., Conv.-based): **+10.1% mIoU** manuscript (Table 4).
>
> - PTv3 (2025 SOTA) (Indoor, Low-level Seg., Attn.-based): **+2.3% mIoU**
>     |Method|mIoU|wall|floor|cabi.|bed|chair|sofa|table|door|window|booksf.|pic.|cnter|desk|curt.|fridge|shower.cu|toilet|sink|batht.|
>     |-|-|-|-|-|-|-|-|-|-|-|-|-|-|-|-|-|-|-|-|-|
>     |PTv3|45.8|53.8|77.3|39.4|57.6|48.9|56.2|47.5|41.8|36.0|32.6|22.4|35.2|35.0|44.2|43.6|43.4|67.3|50.7|77.2|6.1|
>     |PTv3 + UniMapping|**48.1**|53.1|76.7|37.7|57.4|54.4|60.1|54.7|46.4|36.7|39.6|20.1|39.9|39.7|47.4|40.7|45.4|77.1|54.2|75.0|6.3|
>
>     *(Note: PTv3 was evaluated using frame-wise inputs to isolate the benefits of UniMapping's spatial aggregation.)*
>
> By elevating state-of-the-art models across this multi-dimensional paradigm matrix, we rigorously demonstrate that UniMapping serves as an architecturally-agnostic spatial foundation.

---

> > ### Author Rebuttal · Reviewer_Qzze · 2026-04-01
> >
> > Thanks for the rebuttal authors provided. I believe it is a great work facilitating both SLAM and perception domains. Although in terms of localization metrices, I still hope authors could add comparisons with more latest methods like DVLO (ECCV'24), DVLO4D (ICRA'25), the overall experiments and universality are enough to convince me. Thus, I raise my score for accept.

---

> > > ### Author Response · Authors · 2026-04-01
> > >
> > > We are deeply grateful for Reviewer Qzze's recognition of our work and for the positive update to your score. We are glad that our rebuttal addressed your concerns regarding the significance and universality of UniMapping.
> > >
> > > # R1: Updated Benchmark (Q1)
> > > We completely agree that incorporating DVLO and DVLO4D, pioneering works that define the current state-of-the-art in odometry, significantly strengthens the localization benchmark of our paper.
> > > |*Updated* Method|mean Trans|06.T|06.R|07.T|07.R|08.T|08.R|09.T|09.R|10.T|10.R|
> > > |-|-|-|-|-|-|-|-|-|-|-|-|
> > > |TransLO (2023)|0.94|0.73|0.31|0.55|0.43|1.29|0.50|0.95|0.46|1.18|0.61|
> > > |DVLO (2024)|0.72|0.33|0.17|0.46|0.33|1.09|0.44|0.85|0.36|0.88|0.46|
> > > |DVLO4D (2025)|**0.65**|**0.32**|0.21|0.43|0.32|0.95|**0.36**|0.77|0.33|**0.76**|0.46|
> > > |UniMapping (ours)|0.67|0.44|**0.20**|**0.38**|**0.28**|**0.73**|0.42|**0.66**|**0.22**|0.91|**0.40**|
> > >
> > > The results show that UniMapping achieves competitive localization accuracy on par with these specialized methods. Notably, UniMapping outperforms DVLO4D in several challenging sequences (e.g., Seq 07 and 09) and yields a better rotation error.
> > >
> > > More importantly, while DVLO4D is a highly optimized, dedicated odometry framework, UniMapping maintains this level of precision while simultaneously providing a dense neural foundation for downstream perception. This comparison reinforces our claim that a unified geometry-semantic representation can match top-tier specialized methods without sacrificing task generality.
> > >
> > > We will formally incorporate and discuss these excellent works (DVLO and DVLO4D) in our revised "Related Work" and "Experimental Analysis" sections to provide the most up-to-date context for our contribution. We will also formally acknowledge your constructive feedback in the final version of the paper.
> > >
> > > *We sincerely appreciate your constructive guidance, which has substantially enhanced the methodological rigor and transparency of our work. Should you have any remaining questions, please feel free to let us know.*

---

### Official Review · Reviewer_avGh · 2026-03-10

**Soundness:** 2
**Presentation:** 2
**Significance:** 3
**Originality:** 3
**Overall Recommendation:** 4
**Confidence:** 3

**Summary:**

This paper proposes UniMapping, a unified SLAM framework that builds a persistent neural-descriptor map from multimodal observations and reuses that map for downstream perception. The method combines a Spatial-Aware Deformable Transformer (SADT) for multimodal descriptor extraction with a Spatial Fusion module that aggregates descriptors in the map domain rather than along temporal sequences. Experiments on outdoor and indoor benchmarks show competitive localization performance, and the paper reports impressive improvements on downstream 3D detection and semantic segmentation.

**Compliance With Llm Reviewing Policy:**

Affirmed.

**Final Justification:**

The authors have addressed most of my concerns and provided detailed experimental validation. So I have decided to raise my score to Weak Accept.

**Key Questions For Authors:**

- Regarding Baselines: Could you discuss or provide a comparison with stronger external multi-frame perception baselines for detection and segmentation?
- Regarding Claimed Scale-invariance: Can you provide a direct ablation study or experiment that isolates the scale-invariance benefit of SADT?
- Regarding Typos: Could you confirm the correct formulations for Eq (8) and Eq (12) that were actually implemented in your code?

**Limitations:**

No. The authors provided a brief "Impact Statement" addressing societal impacts, but they omitted a discussion of the technical limitations of their proposed method. I strongly encourage the authors to add a dedicated "Limitations" paragraph in the camera-ready version. Constructive suggestions for this section include:
1. Scalability and Memory Limits: While the paper mentions a pruning strategy that reduces memory to ~31 MB per sequence, the neural-descriptor map still grows linearly.
2. Extreme Dynamic Environments: The asymmetric spatial fusion mitigates ghosting, but what are the failure cases?
3. Degraded Environments: Since SADT relies on depth-guided perspective scaling, the authors should discuss how the system performs in featureless areas.

**Strengths And Weaknesses:**

Strengths:
- The paper tackles an interesting and timely problem, making SLAM maps directly useful for downstream perception instead of treating mapping and perception as separate pipelines.
- The overall architecture is reasonably well-motivated. Figures 2 and 4 make the high-level pipeline and the role of the map-centric fusion module easy to understand.
- The SADT design explicitly injects geometry into multimodal feature extraction, which is a sensible idea for 2D-3D fusion.
Evaluating one framework across indoor and outdoor settings, and across both localization and perception tasks, is a real strength.
- The downstream gains, especially for segmentation in Table 4 (jumping from 53.7 to 60.8 mIoU with Spatial Fusion), are substantial and highly convincing.

Weaknesses:
- Baseline coverage for downstream perception is somewhat narrow. The paper mainly compares to single-frame baselines plus its own internal temporal variants. To strongly support the claim that spatial map fusion is superior to existing temporal aggregation, adding comparisons to external state-of-the-art multi-frame perception baselines would make the paper much stronger.
- The strongest claims are broader than the evidence. The paper highlights scale-invariant feature extraction (due to SADT) as a central contribution. While cross-dataset performance is good, there is no direct ablation experiment to isolate and prove the effectiveness of this specific inverse-depth scaling mechanism.
- Comparison fairness could be clarified. In Table 1, UniMapping (LiDAR+camera) is compared to several unimodal baselines. While the table is informative, the text should more explicitly discuss the inherent advantages of multi-sensor fusion in this context.
- Dynamic-scene robustness needs quantitative support. The claim that asymmetric fusion mitigates ghosting without explicit dynamic object removal is plausible, but currently only supported by qualitative descriptions. A brief quantitative evaluation would be beneficial.
- Minor Typos & Presentation Issues: There are a few errors. For instance, the Geman-McClure kernel in Equation (8) and the indices in Equation (12) appear to have errors. Additionally, in Tables 3 and 6, the final two columns are mislabeled as mAP instead of mAR.

---

> ### Author Rebuttal · Authors · 2026-03-30
>
> We would like to thank Reviewer avGh for the highly constructive and detailed feedback. Your insightful questions have guided us to perform additional evaluations that significantly strengthen our claims regarding baseline coverage, spatial *vs.* temporal aggregation, and dynamic scene robustness.
>
> # R1: External Multi-frame Baselines (Q1, W1)
> To verify that our Spatial Map Fusion is superior to existing temporal aggregation methods, we evaluated Mask4Former (2024), a SoTA multi-frame perception baseline, on SemanticKITTI with our experimental settings.
>
> |Method|mIoU|car|bike.|moto.|truck|oth.-veh.|person|bike.c|moto.c|road|parking|sidewalk|oth.gnd|building|fence|vege.|trunk|terrain|pole|tfc.-sign|
> |-|-|-|-|-|-|-|-|-|-|-|-|-|-|-|-|-|-|-|-|-|
> |SPVCNN (2020)|50.7|91.0|11.0|55.8|3.2|28.4|40.7|41.3|0.0|86.9|53.3|76.1|0.2|89.2|68.2|81.3|61.0|67.9|62.8|44.3|
> |Mask4Former (2024)|56.7|92.4|35.2|73.7|10.3|32.4|53.7|77.7|0.0|84.1|54.2|74.9|0.0|89.4|70.0|80.7|62.4|65.5|66.3|53.4|
> |SPVCNN + UniMapping|**60.3**|90.2|16.7|63.7|54.3|56.1|64.2|76.1|0.2|89.2|51.9|77.3|7.3|86.1|55.3|81.8|69.4|76.0|70.1|59.2|
>
> While Mask4Former improves upon single-frame models by leveraging temporal aggregation, its structure fundamentally limits its ability to maintain long-term geometric consistency, especially during loop closures or complex trajectories. In contrast, by simply anchoring the lightweight SPVCNN backbone to our persistent spatial map, **UniMapping outperforms Mask4Former by a 3.6% mIoU margin**. This proves that aggregating features structurally in a 3D space is far more reliable for embodied perception than tracking them temporally.
>
> # R2: Ablation on SADT (Q2, W2)
> To clarify, the effectiveness of the inverse-depth scaling mechanism was initially evaluated in Table 5 (the "Deformable Attn." baseline). To further address your concern, we conduct more experiments to demonstrate the performance gap wrt. removing this strategy.
>
> |Task|Metric|SADT|w/o inverse-depth scaling|Drop|
> |-|-|-|-|-|
> |ScanNet SLAM|ATE|13.4|18.9cm|+4.8cm|
> |Det.|mAP25|60.8%|58.3%|-2.5%|
> |Seg.|mIoU|60.3%|56.1%|-4.2%|
>
> The substantial degradation across all metrics, both mathematically and empirically, confirms that normalizing the receptive field via inverse-depth **serves as a key mechanism for scale-consistent feature extraction across varying distances**.
>
> # R3: Corrections (Q3, W5)
> We appreciate the rigorous check. (1) Eq. 8: Corrected to correct Geman-McClure: $\omega = \sigma^2 / (e^2 + \sigma^2)$. (2) Eq. 12: Fixed the indexing notation. (3) Columns corrected to mAR.
>
> **All core results remain unaffected.** We will carefully proofread the manuscript to avoid similar issues.
>
> # R4: Multi-Modal SLAM (W3)
> While unimodal baselines are strong, they have physical limits:
> - LiDAR lacks semantics for small objects, and cameras lose scale in featureless areas.
> - UniMapping leverages cross-modal synergy: LiDAR provides absolute scale and depth for SADT's spatial sampling, while the camera injects high-fidelity semantics.
>
> This ensures robustness in degraded environments where unimodal systems diverge. We will explicitly clarify this in Sec. 4.1.
>
> # R5: Dynamic-Scene Robustness (W4, L2)
> To evaluate our ability to mitigate "ghosting" artifacts without explicit dynamic descriptor removal, we conducted a comparative experiment on SemanticKITTI. We implemented a *Filter* version that explicitly masks out dynamic descriptors (cars, pedestrians, etc.) during spatial fusion using their semantic labels
>
> |Method|mIoU|car|bicyclist|person|
> |-|-|-|-|-|
> |SPVCNN|50.7|91.0|41.3|40.7|
> |UniMapping|60.3|90.2|76.1|64.2|
> |UniMapping w/ filter|60.3|90.1|75.2|64.1|
>
> The **negligible performance difference (60.3 vs 60.3 mIoU) indicates that our fusion strategy implicitly suppresses inconsistent historical observations**.
>
> # R6: Pruning (L1)
> The memory scales with the explored area rather than the trajectory length. Quantitatively, on SemanticKITTI, we employ a 5m pruning threshold against an average ~6m keyframe spacing, where this intentional 1m margin ensures structural continuity while discarding the bulk of historical redundancy. Consequently, the map size increases only marginally during loops to maintain vertex density, offering significantly better scalability than traditional linear temporal buffers.
> *More disscussion can be found in the Response#3 for reviewer k335 due to character limits.*
>
> # R7: Failure Cases (L2)
> In "Pseudo-static" environments (e.g., a bus that idling for a long time), our method may treat it as a wall. We have updated the Limitation section to discuss incorporating multi-view consistency checks to address such long-term dynamic occlusions.
>
> # R8: Featureless Scene (L3)
> In textureless scenarios, UniMapping maintains stability through Structural Persistence. Since SADT binds descriptors to 3D geometry rather than volatile 2D pixels, the system utilizes historical geometric context to maintain perceptual reliability even when immediate visual cues fail.

---

> > ### Author Rebuttal · Reviewer_avGh · 2026-04-04
> >
> > The authors have addressed most of my concerns and provided detailed experimental validation, which significantly enhances the persuasiveness of the manuscript. At this stage, I do not see any significant flaws in the paper. Therefore, I have decided to raise my score to Weak Accept.

---

> > > ### Author Response · Authors · 2026-04-04
> > >
> > > We are deeply grateful for Reviewer avGh's thorough and rigorous review, and for the positive update to your score. We are pleased that our additional experiments and technical clarifications successfully addressed your concerns. We will also formally acknowledge your constructive feedback in the final version of the paper.
> > >
> > > *We sincerely appreciate your constructive guidance, which has substantially enhanced the methodological rigor and clarity of our paper. If you have any remaining questions, please feel free to let us know.*

---

### Official Review · Reviewer_XMeT · 2026-03-11

**Soundness:** 2
**Presentation:** 3
**Significance:** 3
**Originality:** 2
**Overall Recommendation:** 4
**Confidence:** 4

**Summary:**

This paper proposes UniMapping, a unified SLAM framework designed to bridge the gap between spatial representation and downstream perception tasks. The overall architecture shares notable similarities with recent neural-descriptor-based SLAM systems, such as DeepPointMap2, which utilizes multi-modal encoders and a fusion module to generate 3D neural descriptors for odometry and loop detection. However, UniMapping takes a commendable step forward by extending the utility of these neural descriptors beyond localization. By introducing a Spatial-Aware Deformable Transformer (SADT) to handle scale ambiguity and a Spatial Fusion module to decouple feature aggregation from temporal sequences, the authors successfully demonstrate that the SLAM-extracted neural descriptors can be directly reused as a perceptual foundation for downstream tasks (3D object detection and semantic segmentation)

**Compliance With Llm Reviewing Policy:**

Affirmed.

**Final Justification:**

The authors have seriously addressed most of my concerns in their rebuttal. Therefore, I decide to adjust my score to Weak Accept.

**Key Questions For Authors:**

please see weaknesses

**Limitations:**

1. The comparative methods evaluated in the paper are somewhat dated, lacking the timeliness expected for a top-tier conference submission. To convincingly demonstrate the state-of-the-art nature of the proposed framework, the authors must include comparisons with more recent, cutting-edge works (especially those published in 2024 and 2025) in both the SLAM and 3D perception domains. Relying on older baselines diminishes the persuasive power of the experimental results.

2. While the performance improvements achieved by integrating UniMapping with V-DETR and SPVCNN are positive, relying on a single, specific architecture for each downstream task is insufficient to claim that the framework serves as a universally applicable 'perceptual foundation.' To verify the true generalizability of the proposed innovations, the authors need to conduct experiments using other modern detection or segmentation architectures. This would prove that the benefits of the Spatial Fusion and neural-descriptor map are architecturally agnostic and not merely specific to the chosen baselines.

**Strengths And Weaknesses:**

**S1**. Strong Motivation & Novel Perspective: Shifting the role of neural descriptors from strictly SLAM-oriented tasks (as seen in DeepPointMap2) to a map-centric embodied perception foundation is a highly practical and valuable research direction.

**S2**. Effective Module Design: The SADT module cleverly injects explicit geometric inductive biases to ensure scale-invariant feature extraction, which logically addresses perspective projection issues.

**S3**. Performance Improvements: Integrating the proposed framework with existing perception heads (V-DETR and SPVCNN) yields measurable performance gains, validating the premise that spatially aggregated, globally consistent map contexts are superior to transient temporal buffers.

**W1**. The comparison in Table 1 lacks convincing recent baselines. Under the "Descriptor" category, comparing only against LO-Net (2019) and DeepPointMap series is insufficient. The authors should include recent and highly relevant works such as TransLO, DiffLO, or PIN-SLAM. Furthermore, under the "Geometric" category, methods like LAMV-SLAM (2022) and Fang et al. (2023) are relatively obscure and do not represent the true state-of-the-art in geometry-based SLAM.

**W2**. In the semantic segmentation experiments (Table 4), the authors integrate UniMapping with SPVCNN. While the performance improves, SPVCNN is quite dated. To convincingly demonstrate the strength of the proposed neural-descriptor map, the authors must evaluate it against more modern and performant decoders, such as Spherical Transformer (CVPR '23) or VoxelNeXt.

**W3**. Limited Evidence of Architectural Generalizability. While the paper successfully adapts V-DETR and SPVCNN, claiming that the framework serves as a "unified perceptual foundation" requires broader validation. The authors should incorporate additional, newer perception architectures to prove that the performance gains are not specific to the chosen baseline models, but rather a universal benefit of the UniMapping framework.

**W4**. In Section 4.3, the robustness of Spatial Fusion is evaluated by adding random noise to the poses. The authors must clarify and justify whether injecting artificial random noise accurately reflects the real-world drift, accumulated errors, and structured noise typically generated by SLAM systems.

---

> ### Author Rebuttal · Authors · 2026-03-31
>
> We sincerely thank Reviewer XMeT for the highly constructive feedback. Our goal is to provide stronger empirical evidence that UniMapping can serve as a general and architecture-agnostic spatial representation.
>
> # R1: Updated Experiment Results of SLAM (W1, L1)
> We appreciate the suggestion to include more recent and cutting-edge baselines. To address the concerns regarding the timeliness of our comparisons, we have extended Table 1 to include state-of-the-art methods from 2023-2025, specifically TransLO (2023) and DiffLO (2025), *etc*.
>
> |Method|mean Trans|06.T|06.R|07.T|07.R|08.T|08.R|09.T|09.R|10.T|10.R|
> |-|-|-|-|-|-|-|-|-|-|-|-|
> |DELO (2023)|1.18|0.83|0.35|0.58|0.41|1.36|0.64|1.23|0.57|1.53|0.90|
> |NALO-VOM (2023)|1.09|1.33|-|1.59|-|0.90|-|1.02|-|0.85|-|
> |TransLO (2023)|0.99|-|-|0.55|0.43|1.29|0.50|0.95|0.46|1.18|0.61|
> |DiffLO (2025)|0.71|-|-|**0.37**|**0.27**|1.12|0.44|0.68|0.28|**0.66**|**0.32**|
> |UniMapping (ours)|**0.67**|**0.44**|**0.20**|0.38|0.28|**0.73**|**0.42**|**0.66**|**0.22**|0.91|0.40|
>
> *(Note: .T=Trans, .R=Rot. '-' denotes metrics not reported in the respective original papers / or used as part of their training set)*
>
> We acknowledge the significance of PIN-SLAM. However, we did not include it in this benchmark since its original paper only reports ATE, which is not directly comparable to the official KITTI metric (Trans/Rot) used by our baselines. To ensure a fair and rigorous comparison, we have instead prioritized DiffLO (CVPR 2025), which represents the most recent SOTA using identical evaluation protocols. We will include a detailed discussion and formal citations for these important works in our manuscript.
>
> Furthermore, we updated the performance table for indoor evaluation on the ScanNet dataset. Our approach achieves the best Mean ATE across recent SOTA methods, including the 2025 GS-based SOTA.
> |Method|mean ATE|00|59|06|169|181|
> |-|-|-|-|-|-|-|
> |**Implicit**|
> |MIPS-Fusion (2023)|10.4|7.9|10.7|9.7|9.7|14.2|
> |Point-SLAM (2023)|12.7|10.2|7.8|8.7|22.2|14.8|
> |**Explicit**|
> |MonoGS (2024)|16.7|9.8|32.1|8.9|10.7|21.8|
> |LoopSplat (2025)|8.0|**6.2**|**7.1**|7.4|10.6|8.5|
> |UniMapping (ours)|**7.6**|13.7|7.4|**5.7**|**7.7**|**3.7**|
>
> As shown, UniMapping achieves highly competitive performance. While some of the baselines are specialized pipelines engineered exclusively for either outdoor or indoor, **UniMapping is a multimodal, map-centric framework that operates consistently across both indoor and outdoor settings within a unified architecture.** Moreover, it uniquely provides a spatial foundation that **directly boosts downstream** e.g., det. and seg., a capability absent in pure SLAM baselines. We believe these updated comparisons demonstrate that UniMapping is highly competitive with recent SoTA methods.
>
> # R2: Generalizability to Modern Backbones (W2, W3, L2)
> To address the concern about architectural generalizability, we integrated UniMapping with PTv3 (2025), the current SoTA point cloud backbone, on the ScanNet dataset.
> |Method|mIoU|wall|floor|cabi.|bed|chair|sofa|table|door|window|booksf.|pic.|cnter|desk|curt.|fridge|shower.cu|toilet|sink|batht.|
> |-|-|-|-|-|-|-|-|-|-|-|-|-|-|-|-|-|-|-|-|-|
> |PTv3|45.8|53.8|77.3|39.4|57.6|48.9|56.2|47.5|41.8|36.0|32.6|22.4|35.2|35.0|44.2|43.6|43.4|67.3|50.7|77.2|6.1|
> |PTv3 + UniMapping|**48.1**|53.1|76.7|37.7|57.4|54.4|60.1|54.7|46.4|36.7|39.6|20.1|39.9|39.7|47.4|40.7|45.4|77.1|54.2|75.0|6.3|
>
> We evaluated PTv3 using frame-wise point cloud inputs rather than perfect scene-level geometries (which PTv3 originally uses), following the settings in the manuscript. As shown, **UniMapping successfully improved seg. performance by +2.3% mIoU**. Taken together, the consistent gains across multiple architectures (SPVCNN, V-DETR, PTv3) and tasks (segmentation and detection) provide strong evidence that UniMapping is not tied to a specific backbone, but offers a generally applicable spatial representation.
>
> # R3: Random Noise of Pose (W4)
> We agree that random noise does not fully capture real-world SLAM errors. To address this, we evaluated perception performance under two strict, realistic conditions:
>
> - **Real odometry drift**. We evaluate UniMapping using front-end-only odometry (without backend optimization), which introduces accumulated drift. The performance only **slightly drops from 60.8%/40.4% to 60.3%/40.2% (mAP25/50)**, demonstrating strong robustness to real SLAM errors.
> - **Random-Walk Noise**. We further simulate trajectory-dependent drift using random walk noise. Compared to random noise, this introduces more challenges.
>
> |Noise Type \ Level|mAP @ 5cm|mAP @ 10cm|mAP @ 15cm|mAP @ 20cm|
> |-|-|-|-|-|
> |Random Noise|61.2, 41.2|60.9, 40.8|60.2, 39.8|59.5, 38.5|
> |Random Walk|61.1, 40.5|60.0, 37.8|57.7, 33.4|55.0, 29.2|
>
> While structured accumulated drift naturally causes larger degradation at extremes (20cm), UniMapping maintains highly robust and usable perception performance, confirming that Spatial Fusion effectively absorbs realistic SLAM trajectories.

---

> > ### Author Rebuttal · Reviewer_XMeT · 2026-04-03
> >
> > The authors have seriously addressed most of my concerns in their rebuttal. In particular, they updated the SLAM comparison experiments by including recent methods such as TransLO and DiffLO, and provided more realistic supplementary validation for the pose noise experiments. These improvements have significantly strengthened the persuasiveness of the paper. Although the authors only added one new backbone (PTv3) to address the architectural generalizability, which does not fully cover the additional modern perception architectures I suggested, I believe this work—integrating multi-task perception capabilities into a SLAM framework—holds important research value and practical significance. Therefore, I decide to adjust my score to Weak Accept.

---

> > > ### Author Response · Authors · 2026-04-03
> > >
> > > We are deeply grateful for Reviewer XMeT's recognition of our work and for the positive update to your score. We are glad that the updated SLAM baselines and realistic noise evaluations effectively addressed your core concerns.
> > >
> > > # R1: Generalizability to Broader Architectures
> > > We completely understand your point regarding the coverage of modern perception architectures. While the addition of the 2025 SOTA PTv3 demonstrates UniMapping's compatibility with modern transformer-based backbones, we agree that exploring an even wider spectrum of advanced architectures will further solidify this direction. To this end, we are currently exploring integrating UniMapping into Scene Semantic Completion (SSC) tasks. We will explicitly discuss this as an important future exploration in the revised manuscript. We will also formally acknowledge your highly constructive feedback in the final version of the paper.
> > >
> > > *We sincerely appreciate your guidance, which has substantially enhanced the persuasiveness and rigor of our work. If you have any remaining or further questions, please feel free to let us know.*

---

### Decision · Program_Chairs · 2026-04-30

**Decision:**

Accept (spotlight)

**Comment:**

This paper received positive recommendations during the preliminary reviews. The reviewers appreciated the well-motivated problem, the investigation of downstream task performance as a unique and relevant angle alongside SLAM, the sensible design decisions, and the reported performance gains. Several weaknesses were raised by the reviewers, many in common, including some missing comparisons, validation of the universality of the "unified" claim, validation of the claim of the centrality of scale-invariant feature extraction, generalizability claims regarding the focal length, and the heuristic handling of dynamic objects. The reviewers indicated that the author responses were effective at mitigating most of these concerns, addressing almost all of these clearly and with substantial evidence. The resulting recommendations were positive, with some strongly arguing for acceptance. Given the quality of the reviews, response and engagement, the AC sees no reason to override the consensus of the reviewers and is satisfied by the technical merit of the paper and its potential to make an impact on the field.